# β-arrestin 2 as an activator of cGAS-STING signaling and target of viral immune evasion

Yihua Zhang[1,5], Manman Li[1,5], Liuyan Li[1], Gui Qian[1], Yu Wang[2], Zijuan Chen[1], Jing Liu[1], Chao Fang[3], Feng Huang[4], Daqiao Guo[3], Quanming Zou[2], Yiwei Chu[1] & Dapeng Yan [1✉]

Virus infection may induce excessive interferon (IFN) responses that can lead to host tissue injury or even death. β-arrestin 2 regulates multiple cellular events through the G protein-coupled receptor (GPCR) signaling pathways. Here we demonstrate that β-arrestin 2 also promotes virus-induced production of IFN-β and clearance of viruses in macrophages. β-arrestin 2 interacts with cyclic GMP-AMP synthase (cGAS) and increases the binding of dsDNA to cGAS to enhance cyclic GMP-AMP (cGAMP) production and the downstream stimulator of interferon genes (STING) and innate immune responses. Mechanistically, deacetylation of β-arrestin 2 at Lys171 facilitates the activation of the cGAS–STING signaling and the production of IFN-β. In vitro, viral infection induces the degradation of β-arrestin 2 to facilitate immune evasion, while a β-blocker, carvedilol, rescues β-arrestin 2 expression to maintain the antiviral immune response. Our results thus identify a viral immune-evasion pathway via the degradation of β-arrestin 2, and also hint that carvedilol, approved for treating heart failure, can potentially be repurposed as an antiviral drug candidate.

[1] Department of Immunology, School of Basic Medical Sciences & Shanghai Public Health Clinical Center, Key Laboratory of Medical Molecular Virology of MOE/MOH, Fudan University, Shanghai 200032, China. [2] Department of Microbiology and Biochemical Pharmacy, National Engineering Research Centre of Immunological Products, College of Pharmacy, Army Medical University, Chongqing 400038, China. [3] Department of Vascular Surgery, Zhongshan Hospital Affiliated to Fudan University, Institute of Vascular Surgery, Fudan University, Shanghai 200032, China. [4] Beijing Hospital of Traditional Chinese Medicine, Capital Medical University, Beijing 100010, China. [5] These authors contributed equally: Yihua Zhang, Manman Li. ✉email: dapengyan@fudan.edu.cn

Viruses are worldwide issues that threaten human health and lead to huge economic losses and social problems. The emergence of the novel coronavirus, SARS-CoV-2, has resulted in a tremendous public health crisis in China that is also having an impact at a global level[1–3]. As the first line of immune response, the innate immunity is activated by recognizing pathogen-associated molecular patterns (PAMPs) through a set of pattern recognition receptors (PRRs), which can subsequently regulate the adaptive immunity and together eliminate the viruses[4]. Despite the diversity of viruses, host immune cells are able to evolve various powerful PRRs and innate immune signaling pathways to detect the PAMPs. Toll-like receptors (TLRs), retinoic acid-inducible gene I (RIG-I)-like receptors (RLRs), and DNA sensors recognize viral nucleic acids and initiate downstream signal transduction, resulting in the induction of type I interferons (IFNs) and other proinflammatory cytokines to protect the host against virus invasion[5,6]. The endosomal TLRs recognize viral double-stranded RNA (dsRNA; TLR3), single-stranded RNA (ssRNA; TLR7/8), or unmethylated CpG sequences in the DNA (TLR9)[7–10]. RIG-I and melanoma differentiation-associated gene 5 (MDA5) are known to recognize viral dsRNA[11]. In addition, DNA sensors, such as cyclic GMP-AMP synthase (cGAS), recognize dsDNA[12–14]. TLR3 recruits TIR-domain-containing adapter-inducing interferon-β (TRIF), TNFR-associated factor (TRAF) 2/6, and TANK-binding kinase-1 (TBK1)[15]. RIG-I and MDA5 recruit mitochondrial antiviral-signaling protein (MAVS) to activate TBK1[16]. cGAS synthesizes cGAMP as a second messenger to activate STING and recruit TBK1[17–19]. The TBK1 complex promotes the phosphorylation of IFN regulatory factor 3 (IRF3), leading to the dimerization and translocation of IRF3 from the cytoplasm to the nucleus, which interacts with the interferon-stimulated response elements (ISRE) to produce type I IFNs and proinflammatory cytokines[14].

cGAS specifically recognizes dsDNA and cDNA that are reverse-transcribed from RNA in RNA viruses, such as HIV-1, to produce cGAMP and subsequently activates STING. STING recruits and activates TBK1 and IRF3, and induces the production of type I IFNs and proinflammatory cytokines. The enzymatic function of cGAS is achieved by the C-terminal part, which possesses nucleotidyltransferase activity and MB21 domains along with DNA-binding sites[20]. The interaction between cGAS and viral DNA in the cytosol induces the formation of liquid-like droplets, resulting in the conformational change and oligomerization of cGAS, which promotes the activation of the enzyme activity of cGAS[21,22]. Except for the pivotal role of cGAS in virus infection, cGAS is important for maintaining homeostasis in several cells. Numerous studies have reported that cGAS could be regulated by several posttranslational modifications, including sumoylation, acetylation, and ubiquitination[23–25]. On the other hand, although cGAS acts as a cytosolic DNA sensor that initiates the STING–TBK1–IRF3-type I IFN signaling cascade, the location of cGAS largely influences its functions. cGAS not only diffuses throughout the cytosol to search its DNA ligand but also localizes at the plasma membrane and in the nucleus, which is associated with distinguishing self- and viral DNA, DNA damage, and tumor growth[26–28].

β-arrestin 2 is a multifunctional adaptor that plays a role in the desensitization, internalization, and signaling transduction of different types of cell surface receptors, especially activated GPCRs[29–33], to regulate signal transduction and participate in different diseases[34–37]. Previous studies have demonstrated that USP33 and Mdm2 function reciprocally and favor respectively the stability and lability of the β-arrestin 2 complex[38]. β-arrestin 2 prolongs the activation of ERK through the G protein-dependent pathway to regulate microglia chemotaxis[39]. Conversely, the interaction between β-arrestin 2 and IκB suppresses the phosphorylation and degradation of IκB, which suppresses the activation and release of p65, thus inhibiting the activation of NF-κB signaling[40]. Moreover, β-arrestin 2 interacts with TRAF6 in response to LPS or IL-1β stimulation and prevents the oligomerization and autoubiquitination of TRAF6 to negatively regulate the TLR–IL-1R signaling pathways[41]. The drug carvedilol is used to treat heart failure and hypertension. Mechanistically, carvedilol induces β-adrenergic receptor (βAR)-mediated EGFR transactivation and downstream ERK activation in a β-arrestin-dependent manner[42]. Specifically, carvedilol was found to increase β-arrestin 2 expression and the recruitment of β-arrestin 2 to the β2AR[43,44].

In the present study, we observe that β-arrestin 2 positively regulates the production of IFN-β by targeting cGAS. Meanwhile, we also demonstrate that deacetylation of β-arrestin 2 at Lys171 enhances the antiviral immune response. In addition, carvedilol blocks the degradation of β-arrestin 2, which is induced by virus infection. These findings not only elaborate a novel mechanism of viruses that evade host immune response by degrading the positive regulator β-arrestin 2 but also demonstrate that carvedilol can be a potential antiviral drug candidate by promoting β-arrestin 2 expression.

## Results

**Deficiency of β-arrestin 2 decreases cellular antiviral responses.** To investigate the function of β-arrestin 2 in the antiviral innate immune response, we analyzed β-arrestin 2 expression in peritoneal macrophages after infection with herpes simplex virus 1 (HSV-1) or vesicular stomatitis virus (VSV). We observed that β-arrestin 2 protein level was decreased (Fig. 1a and Supplementary Fig. 1a), whereas the abundance of mRNA expression of β-arrestin 2 was not affected by the virus infection (Fig. 1b), indicating that β-arrestin 2 is probably degraded by viruses in a posttranslational manner.

We then prepared peritoneal macrophages from wild-type (WT) and $Arrb2^{-/-}$ mice and stimulated them with HSV-1, VSV, Sendai virus (SeV), or DNA mimics. The expression of $Ifnb$, $Isg15$, and $Ccl5$ was significantly downregulated in $Arrb2^{-/-}$ macrophages stimulated with all types of stimulations compared with the expression of these genes in their WT counterparts (Fig. 1c, d and Supplementary Fig. 1b–e). Consistent with this finding, the virus titers were much higher in $Arrb2^{-/-}$ macrophages than in WT macrophages (Fig. 1e, f). We next analyzed the activation of the major pathways that are involved in virus infection, including type I IFN, NF-κB, and MAPK pathways. We observed that HSV-1 and VSV infection induced reduced phosphorylation of TBK1, IRF3, IRF7, and STING in $Arrb2^{-/-}$ macrophages than in WT macrophages (Fig. 1g, h). However, the phosphorylation of p65, p38, ERK, and Jnk remained unaffected, indicating that β-arrestin 2 primarily promotes the IFN-β signaling pathway to regulate the antiviral innate immune response. The dimerization and the translocation of IRF3 are essential for the production of IFN-β during virus infection. As expected, the dimerization (Fig. 1i, j) and translocation (Fig. 1k, l and Supplementary Fig. 1f) of IRF3 in $Arrb2^{-/-}$ macrophages were significantly reduced compared with those in WT macrophages.

To further confirm the function of β-arrestin 2, we transfected RAW264.7 cells with the negative control small interfering RNA (siRNA) or β-arrestin 2-specific siRNA and confirmed that only β-arrestin 2-specific siRNA decreased β-arrestin 2 expression (Supplementary Fig. 1g, h). As anticipated, HSV-1, VSV, ISD, and RNA mimics poly(I:C) stimulated much lower expression of $Ifnb$, $Isg15$, and $Ccl5$ in β-arrestin 2 knockdown cells (Fig. 1m and Supplementary Fig. 1i–l). Meanwhile, the titers of HSV-1 and

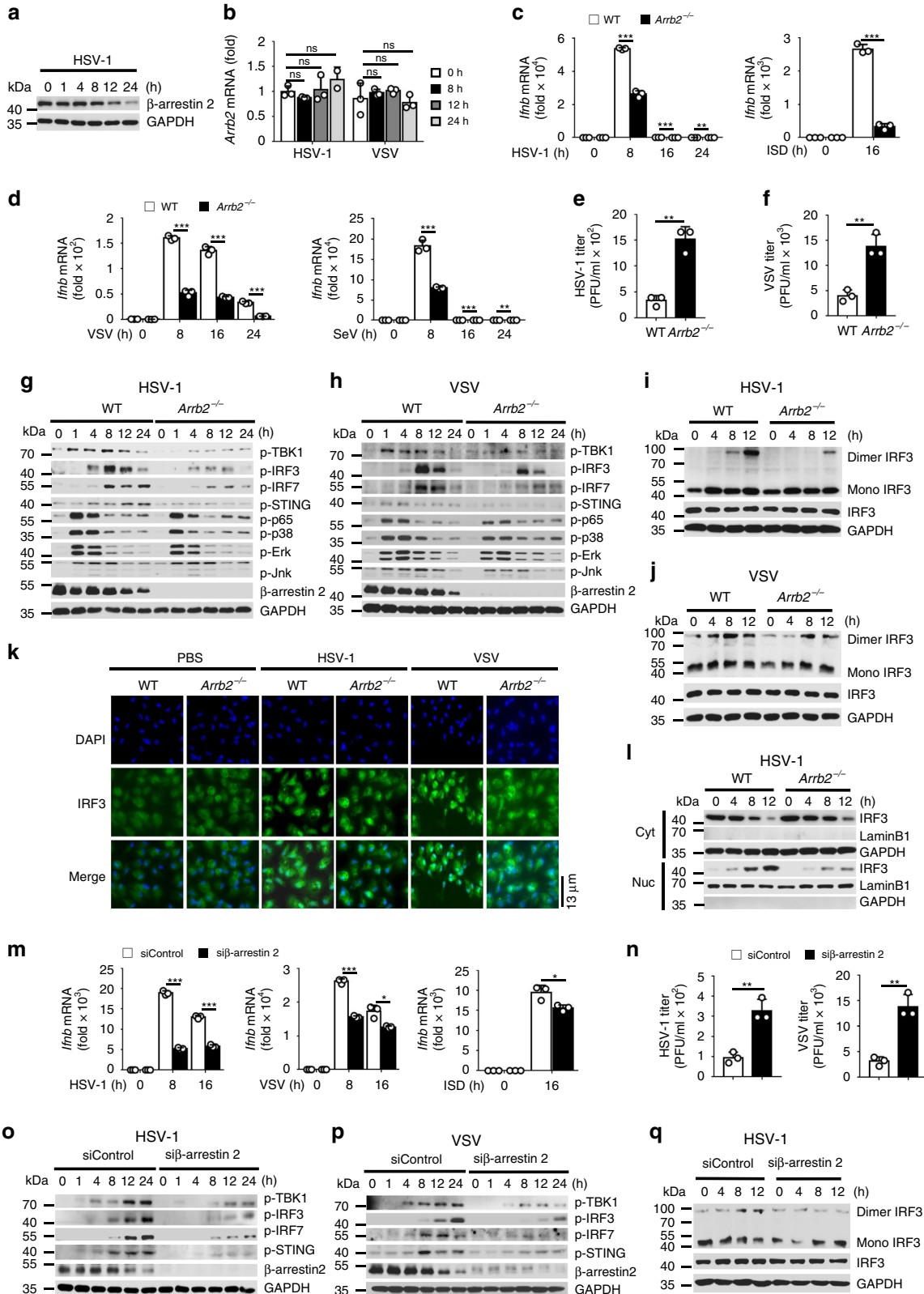

VSV were increased in β-arrestin 2 knockdown cells (Fig. 1n). Consistent with the cytokine levels, the knockdown of β-arrestin 2 significantly reduced the phosphorylation of TBK1, IRF3, IRF7, and STING (Fig. 1o, p) and also the dimerization of IRF3 (Fig. 1q) during virus infection. Altogether, these results suggested that β-arrestin 2 promoted the IFN-β signaling and the antiviral immune response induced by RNA or DNA virus.

**β-arrestin 2 positively regulates IFN-β signaling.** Next, we investigated whether β-arrestin 2 overexpression had a substantial effect on IFN-β signaling. We observed that in L929 mouse fibroblast cells, β-arrestin 2 overexpression significantly increased the HSV-1- or VSV-induced expression of *Ifnb*, *Isg15*, and *Ccl5* (Fig. 2a, b). Similar results were obtained when poly(I:C) or ISD was used as a stimulus (Fig. 2c, d). Consistent with this finding,

**Fig. 1 Deficiency of β-arrestin 2 decreases cellular antiviral responses. a** Immunoblot of lysates of peritoneal macrophages infected with HSV-1 for indicated times. **b** *Arrb2* mRNA levels in peritoneal macrophages infected with HSV-1 or VSV for indicated times. **c, d** *Ifnb* mRNA levels in WT or *Arrb2*⁻/⁻ mouse peritoneal macrophages stimulated with HSV-1 (left, **c**) (***$P < 0.0001$, = 0.0006 in sequence, **$P = 0.0029$), ISD (right, **c**) (***$P < 0.0001$), VSV (left, **d**) (***$P < 0.0001$, < 0.0001, < 0.0001 in sequence), or SeV (right, **d**) (***$P = 0.0003$, 0.0002 in sequence, **$P = 0.0070$) for indicated times. **e, f** The virus titers as in (left, **c**) (***$P = 0.0014$) or (left, **d**) (***$P = 0.0031$) for 72 h. **g, h** Immunoblot of lysates of peritoneal macrophages from WT or *Arrb2*⁻/⁻ mice infected with HSV-1 **g** or VSV **h** for indicated times. **i, j** Immunoblot analysis of monomeric and dimeric IRF3 as in **g** or **h**. **k** Immunofluorescence microscopy of peritoneal macrophages from WT or *Arrb2*⁻/⁻ mice infected with PBS (left), HSV-1 (middle), or VSV (right) for 8 h. **l** Immunoblot analysis of nuclear (Nuc) and cytoplasmic (Cyt) fractions as in **i. m** *Ifnb* mRNA levels in RAW264.7 cells transfected with control siRNA or β-arrestin 2 siRNA and then stimulated with HSV-1 (left), VSV (middle), or ISD (right) for indicated times (***$P < 0.0001$, < 0.0001 in sequence, left panel; ***$P < 0.0001$, *$P = 0.0125$, middle panel; *$P = 0.0239$, right panel). **n** The virus titers as in **m** for 72 h (***$P = 0.0031$, left panel; *$P = 0.0019$, right panel). **o, p** Immunoblot of lysates of RAW264.7 cells treated with indicated siRNA and infected with HSV-1 **o** or VSV **p** for indicated times. **q** Immunoblot analysis of monomeric and dimeric IRF3 as in **o**. ns not significant. Data are representative of at least three independent experiments (mean ± SEM. in **b**–**f**, **m**, **n**, $n = 3$). Two-tailed unpaired Student's *t*-test.

the titers of HSV-1 or VSV were decreased in β-arrestin 2-overexpressing L929 cells (Fig. 2e, f). We next detected the phosphorylation of TBK1, IRF3, IRF7, and STING after HSV-1 or VSV infection. As expected, β-arrestin 2 overexpression in L929 cells induced even higher phosphorylation of TBK1, IRF3, IRF7, and STING than that in the control group (Fig. 2g, h). Furthermore, cells with ectopic β-arrestin 2 expression exhibited significantly elevated dimerization of STING (Fig. 2i) and IRF3 (Fig. 2j) and translocation of IRF3 to the nucleus (Fig. 2k, l) after stimulation with either HSV-1 or VSV. These results indicated that β-arrestin 2 enhanced the translocation of IRF3 from the cytoplasm to the nucleus and IFN-β signaling by promoting the formation of STING and IRF3 dimers.

**Deficiency of β-arrestin 2 diminished the antiviral immune response in vivo**. To investigate the function of β-arrestin 2 in the antiviral innate immune response in vivo, we infected 6-week-old homozygous *Arrb2*⁻/⁻ mice and their WT counterparts with HSV-1 ($2 \times 10^7$ PFU/mouse) or VSV ($5 \times 10^8$ PFU/mouse) and evaluated the expression levels of *Ifnb*, *Isg15*, and *Ccl5* in different tissues. After the infection of mice with HSV-1, the expression levels of *Ifnb*, *Ccl5*, and *Isg15* were significantly lower in the spleens, livers, and lungs of *Arrb2*⁻/⁻ mice than in those of WT mice (Fig. 3a). Consistent with this result, we observed significantly higher HSV-1 titers in the spleens, livers, and lungs of *Arrb2*⁻/⁻ mice than in those of WT mice (Fig. 3b). Similar results were obtained when VSV was used as a stimulus (Fig. 3c, d). In agreement with these results, we observed more injury in the lungs and spleens of *Arrb2*⁻/⁻ mice than in those of WT mice (Fig. 3e, f). We further compared the survival rates between WT and *Arrb2*⁻/⁻ mice after infection with HSV-1 or VSV and observed that *Arrb2*⁻/⁻ mice were more susceptible to HSV-1 or VSV and suffered higher mortality than their WT counterparts (Fig. 3g, h). All these in vivo results indicated that β-arrestin 2 was a critical positive regulator of the antiviral immune response to both RNA and DNA viruses.

**β-arrestin 2 interacts with cGAS and promotes the activation of cGAS**. To further investigate the function of β-arrestin 2 in host antiviral immunity against virus infection, we performed RNA-seq analyses of WT and *Arrb2*⁻/⁻ PMs infected with HSV-1. Compared to WT group, *Arrb2*⁻/⁻ group significantly altered the expression of 859 genes (251 upregulated and 608 downregulated) (Fig. 4a and Supplementary Fig. 2a). The PPI analyses based on differential genes showed a close interaction network among host antiviral genes, such as *Isg15*, *Cxcl10*, *Isg20*, IFIT family genes, and so on (Supplementary Fig. 2b). Consistently, functional classification analysis of the downregulated genes revealed an enrichment in positive regulation genes of host

defense response to virus (Supplementary Fig. 2c). Gene ontology analyses showed that *Arrb2*⁻/⁻ group downregulated the expression of multiple genes related to response to external stimulus, immune response, inflammatory response, innate immune response, response to virus, etc (Fig. 4b and Supplementary Fig. 2d). Similarly, KEGG analyses revealed an enrichment of the differential genes in cytokine–cytokine receptor interaction, chemokine signaling pathway, primary immunodeficiency, Jak–STAT signaling pathway, cytosolic DNA-sensing pathway, and so on (Fig. 4c and Supplementary Fig. 2e). We next aimed to determine the mechanism through which β-arrestin 2 promotes IFN-β signaling. To identify the component in IFN-β signaling that was affected by β-arrestin 2, we performed a dual-luciferase assay through overexpressing β-arrestin 2 in HEK293T cells in the presence or absence of expression vectors for RIG-I-N, MAVS, cGAS plus STING, TBK1, or constitutively active IRF3 (termed as "IRF3-5D" here). Our results revealed that β-arrestin 2 overexpression promoted the activity of luciferase reporters for above signaling proteins-induced IFN-β and ISRE activation except for IRF3-5D (Fig. 4d, e), suggesting that β-arrestin 2 promoted the activation of signaling proteins upstream of IRF3. A co-immunoprecipitation-based screening experiment demonstrated that β-arrestin 2 interacted with only cGAS (Fig. 4f, g), but not with other proteins in the IFN-β signaling pathway. We next used *E. coli* to express and purify GST-cGAS and His-β-arrestin 2. Using these purified proteins in an in vitro GST precipitation assay, we found that cGAS directly bound with β-arrestin 2 (Fig. 4h). The luciferase assay demonstrated that β-arrestin 2 promoted cGAS plus STING-triggered activation of the IFNB and ISRE promoters in a dose-dependent manner (Fig. 4i, j). To further study which domains of cGAS and β-arrestin 2 are involved in their interaction, we constructed a series of truncation mutants of cGAS and β-arrestin 2, respectively. We found, using truncated proteins, that sequence regions 1–185 of β-arrestin 2 and 213–522 of cGAS were important for their interactions (Supplementary Fig. 2f, g).

Recognition of dsDNA by cGAS produces cGAMP and subsequently activates STING, which activates IFN-β signaling transduction[13,14]. As β-arrestin 2 was found to interact with cGAS, we assessed the effect of β-arrestin 2 on the binding of dsDNA and cGAS. We used a pull-down assay to immunoprecipitate biotin-labeled dsDNA with streptavidin and found that β-arrestin 2 significantly increased the amount of cGAS pulled down by dsDNA[28] (Fig. 4k), indicating that β-arrestin 2 improved the DNA-binding ability of cGAS. We also found that β-arrestin 2 could promote cGAS and DNA-binding activity in vitro using purified proteins by EMSA (Fig. 4l). As cGAS forms dimers after dsDNA binding[21,45], we next aimed at investigating whether β-arrestin 2 influenced the formation of cGAS dimers. Our results revealed that β-arrestin 2 overexpression significantly

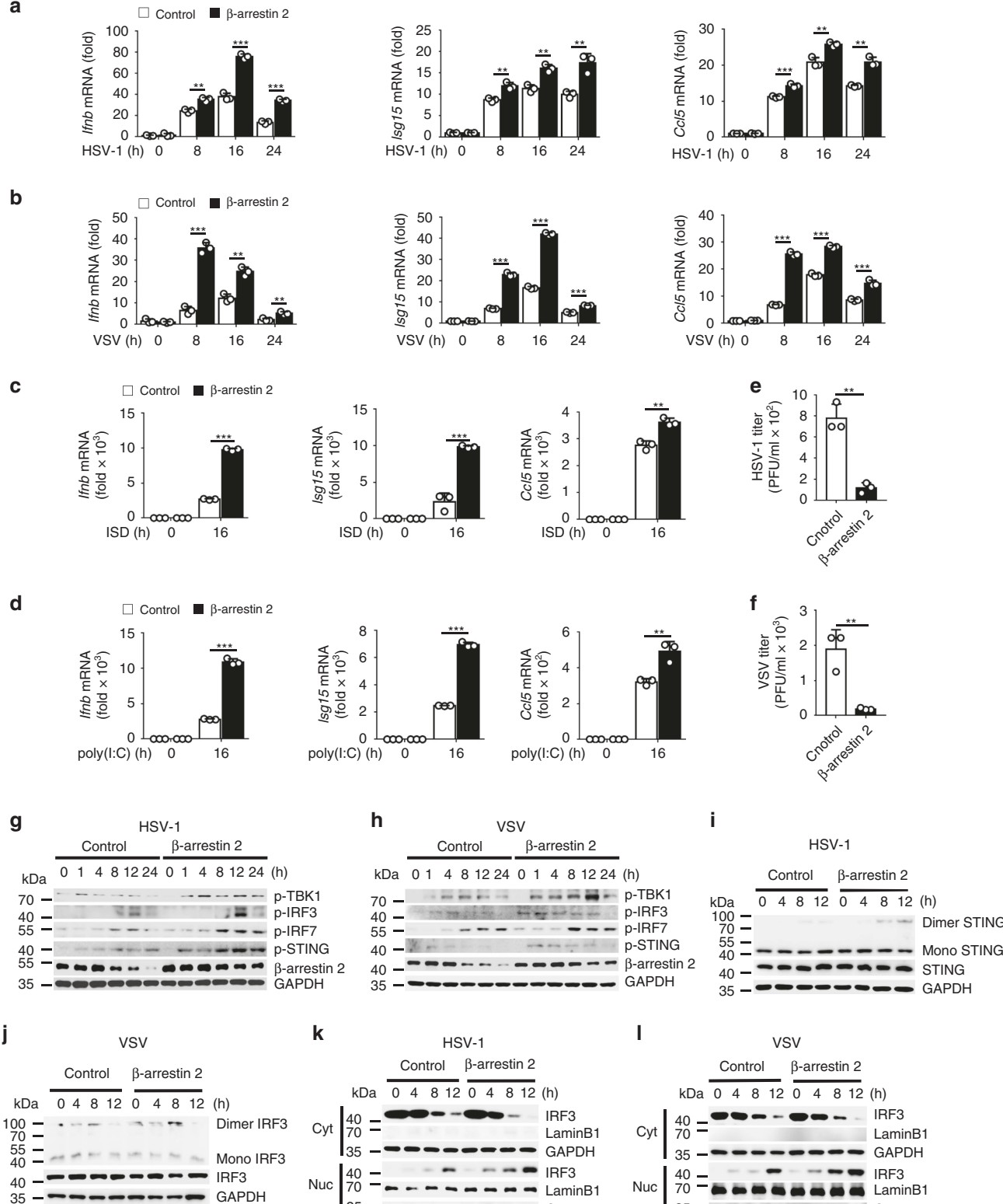

**Fig. 2 β-arrestin 2 positively regulates IFN-β signaling. a–d** *Ifnb*, *Isg15*, and *Ccl5* mRNA levels in L929 cells transfected with control or β-arrestin 2 vectors and stimulated with HSV-1 **a** (**P = 0.0018, ***P < 0.0001, < 0.0001 in sequence, left panel; **P = 0.0040, 0.0028, 0.0050 in sequence, middle panel; ***P = 0.0006, **P = 0.0044, 0.0010 in sequence, right panel), VSV **b** (***P < 0.0001, **P = 0.0015, 0.0071 in sequence, left panel; ***P < 0.0001, < 0.0001, = 0.0001 in sequence, middle panel; ***P < 0.0001, < 0.0001, = 0.0009 in sequence, right panel), ISD **c** (***P < 0.0001, left panel; ***P = 0.0004, middle panel; **P = 0.0025, right panel), or poly(I:C) **d** (***P < 0.0001, left panel; ***P < 0.0001, middle panel; **P = 0.0067, right panel) for indicated times. **e, f** The virus titers as in **a** or **b** for 72 h (**P = 0.0013 in **e**, **P = 0.0062 in **f**). **g, h** Immunoblot of lysates of L929 cells transfected with control or β-arrestin 2 vectors and infected with HSV-1 **g** or VSV **h** for indicated times. **i, j** Immunoblot analysis of monomeric and dimeric STING **i** and IRF3 **j** as in **g** or **h**. **k, l** Immunoblot analysis of nuclear and cytoplasmic fractions as in **i** or **j**. Data are representative of at least three independent experiments (mean ± SEM in **a–f**, n = 3). Two-tailed unpaired Student's *t*-test.

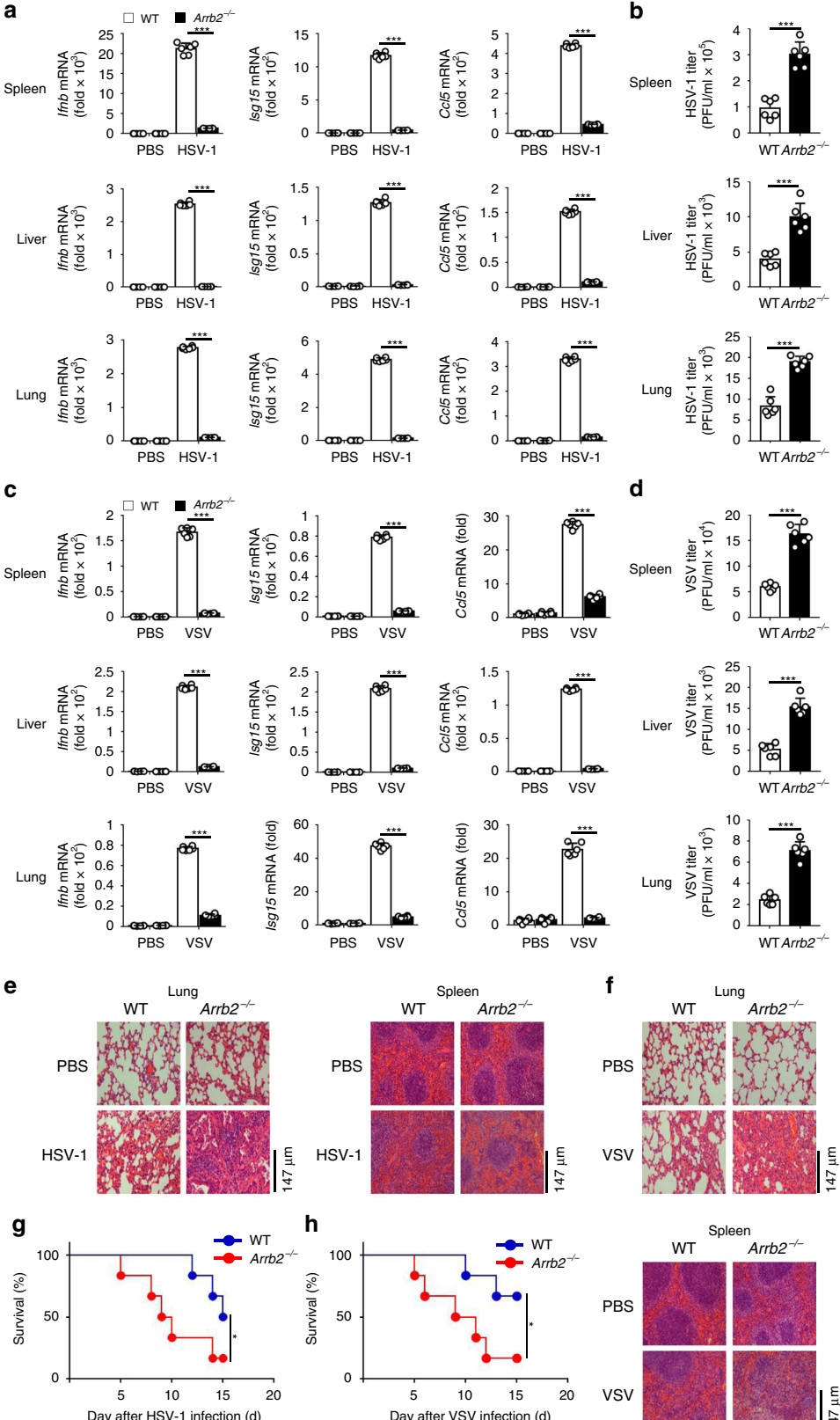

**Fig. 3 Deficiency of β-arrestin 2 diminished the antiviral immune response in vivo. a** *Ifnb*, *Isg15*, and *Ccl5* mRNA levels in the spleens, livers, and lungs of WT and *Arrb2⁻/⁻* 6-week-old male mice (*n* = 6 per group) intraperitoneally injected with PBS or HSV-1 (2 × 10⁷ PFU per mouse) for 48 h (***P < 0.0001 in all panels). **b** The virus titers as in **a** (***P < 0.0001 in all panels). **c** VSV (5 × 10⁷ PFU per mouse) infection as in **a** (***P < 0.0001 in all panels). **d** The virus titers as in **c** (***P < 0.0001 in all panels). **e, f** Microscopy of hematoxylin- and eosin-stained lung and spleen sections as in **a** or **c**. **g, h** Survival of WT and *Arrb2⁻/⁻* 6-week-old male mice (*n* = 6 per group) infected intraperitoneally with a high dose of HSV-1 (2 × 10⁸ PFU per mouse) **g** (*P = 0.0430) or VSV (5 × 10⁸ PFU per mouse) **h** (*P = 0.0411) and monitored for 15 days. Data are representative of at least three independent experiments (mean ± SEM, two-tailed unpaired Student's *t*-test in **a**–**d** or Kaplan–Meier analysis in **g**, **h**).

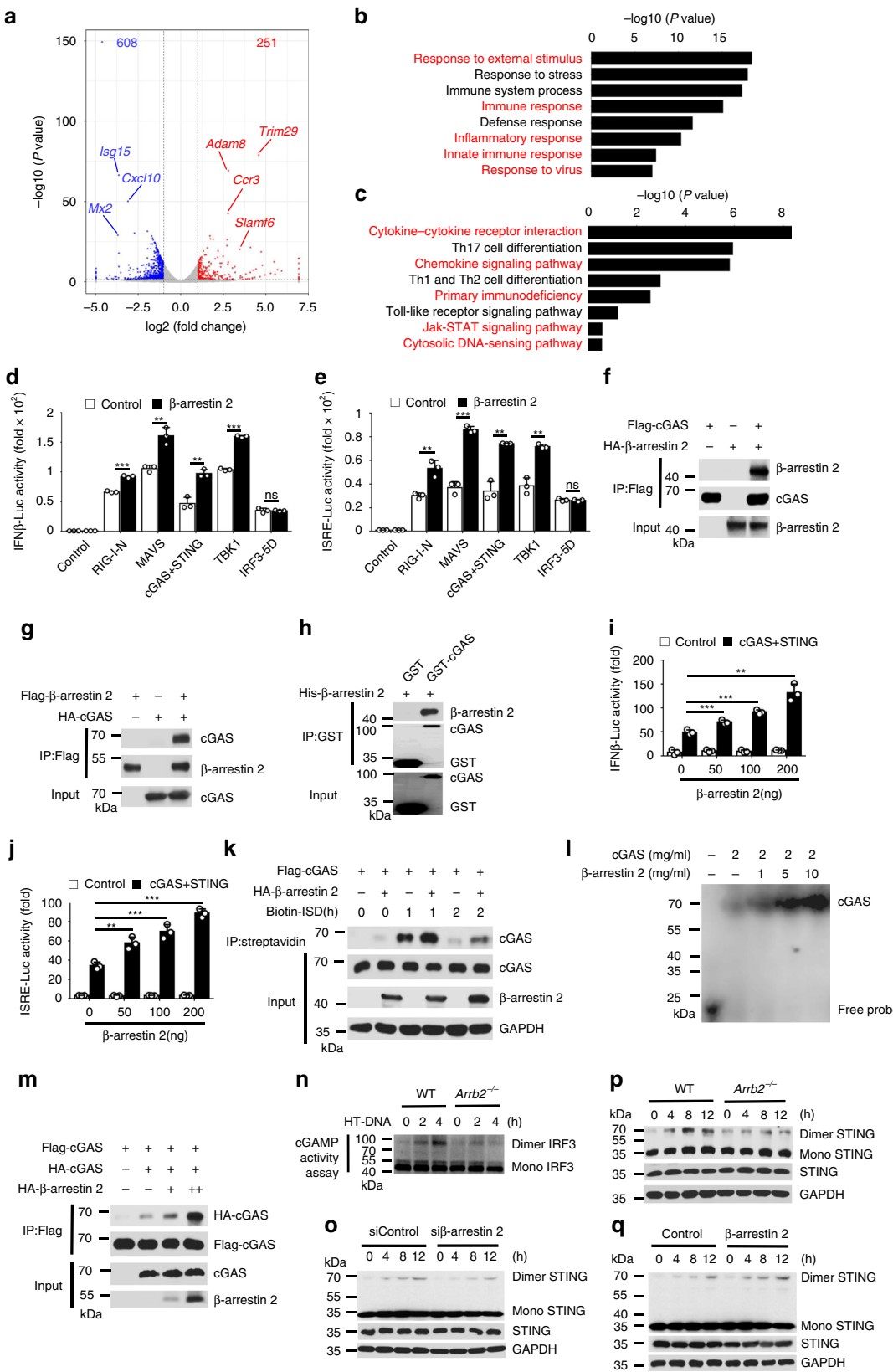

enhanced the interaction between Flag-tagged cGAS and HA-tagged cGAS (Fig. 4m). Moreover, we detected the production of cGAMP measured by the dimerization of IRF3[14]. In fact, extracts from HT-DNA-stimulated *Arrb2*$^{-/-}$ peritoneal macrophages exhibited less cGAMP activity compared with that from WT macrophages, which resulted in reduced IRF3 dimerization in the permeabilized peritoneal macrophages (Fig. 4n), suggesting that β-arrestin 2 deficiency impaired the production of cGAMP. We next examined the dimerization of STING and found that knockdown or knockout of β-arrestin 2 decreased, but β-arrestin 2 overexpression increased the dimerization of STING (Fig. 4o, p, q). Taken together, these findings suggested that β-arrestin 2

**Fig. 4 β-arrestin 2 interacts with cGAS and promotes the activation of cGAS. a** Volcano plots comparing gene expression in wild-type versus $Arrb2^{-/-}$ infected peritoneal macrophages. **b** GO enrichment analysis of differential genes involved in the terms as indicated. **c** KEGG enrichment analysis of differential genes involved in the pathways as indicated. **d, e** Luciferase assay of IFN-β **d** (***$P < 0.0001$, **$P = 0.0033$, $0.0021$ in sequence, ***$P < 0.0001$) and ISRE **e** (**$P = 0.0052$, ***$P = 0.0001$, **$P = 0.0010$, $0.0012$ in sequence) activation in HEK293T cells expressing various vectors. **f, g** Immunoassay of lysates of HEK293T cells expressing various vectors. **h** Direct binding of His–β-arrestin 2 with GST-cGAS. **i, j** Luciferase assay of IFN-β **i** (***$P < 0.0001$, $<0.0001$ in sequence, **$P = 0.0003$) and ISRE **j** (**$P = 0.0001$, ***$P < 0.0001$, $< 0.0001$ in sequence) activation in HEK293T cells expressing various vectors. **k** Immunoassay of cell lysates and streptavidin-precipitated proteins from HEK293T cells transfected with various vectors and stimulated with biotin-ISD. **l** EMSA of cGAS and dsDNA. **m** Immunoassay of lysates of HEK293T cells expressing various vectors. **n** cGAMP activity measured by extracts from HT-DNA-stimulated (6 h) WT and $Arrb2^{-/-}$ peritoneal macrophages. **o–q** Immunoblot analysis of monomeric and dimeric STING in WT and $Arrb2^{-/-}$ peritoneal macrophages **o**, control siRNA or siβ-arrestin 2-transfected PAW264.7 cells **p**, and control or β-arrestin 2 vector-transfected L929 cells **q**. Data are representative of at least three independent experiments (mean ± SEM in **d, e, i, j**, $n = 3$). Two-tailed unpaired Student's $t$-test.

directly interacted with cGAS and promoted IFN-β signaling by enhancing the DNA-binding ability of cGAS.

As β-arrestin 2 also influenced the VSV-induced activation of IFN-β signaling, while only binding with cGAS directly, we thought that there might exist some other proteins that could be associated with β-arrestin 2. β-arrestin 2 did not associate with TBK1 or TRAF3 without virus infection, while the situation changed after infection. We found that endogenous β-arrestin 2 could be associated with TBK1 and TRAF3 during virus infection (Supplementary Fig. 2h). Furthermore, β-arrestin 2 could promote the interaction between TBK1 and TRAF3, which might explain why β-arrestin 2 could promote poly(I:C)- and VSV-induced activation of IFN-β signaling (Supplementary Fig. 2i).

**Deacetylation of β-arrestin 2 at Lys171 enhanced the antiviral immune response.** As β-arrestin 2 expression was decreased after the virus infection, which might be a smart immune evasion strategy by viruses, we conducted mass spectrometry analysis of β-arrestin 2 in the presence or absence of HSV-1 infection. This analysis revealed that the acetylation at Lys171 and Lys227 sites and the ubiquitination at Lys12, Lys18, Lys171, Lys178, and Lys227 sites were changed after HSV-1 infection. Interestingly, the Lys171 site of β-arrestin 2 was acetylated before infection but was ubiquitinated after HSV-1 infection (Supplementary Fig. 3a), suggesting that acetylation and ubiquitination regulated the degradation of β-arrestin 2. We replaced the lysines with arginines (acetylation- and ubiquitination-defective mutants) or glutamines (acetylation-mimetic mutants), overexpressed WT β-arrestin 2 and these mutants in HeLa cells, and then infected them with HSV-1 or VSV to detect the expression of *Ifnb*. We observed that K171R promoted but K171Q inhibited the expression of *Ifnb*, which indicated that Lys171 is a pivotal residue for β-arrestin 2 function (Fig. 5a). Similar results were obtained with the luciferase assay (Fig. 5b). Furthermore, the production of *Ifnb*, *Isg15*, and *Ccl5* was increased in $Arrb2^{-/-}$ mouse embryonic fibroblasts (MEFs) transfected with K171R mutant compared with WT β-arrestin 2, whereas K171Q mutant abolished the positive regulation during virus infection (Fig. 5c and Supplementary Fig. 3b, c). Consistent with this observation, K171R mutant inhibited but K171Q mutant promoted the virus titers (Fig. 5d). Furthermore, K171R mutant enhanced the phosphorylation of TBK1, IRF3, IRF7, and STING and the dimerization and translocation of IRF3, compared with WT β-arrestin 2, whereas K171Q mutant abolished these functions (Fig. 5e–g, and Supplementary Fig. 3d, e, f). These findings suggested that the deacetylation of β-arrestin 2 at Lys171 was necessary for positive regulation of the IFN-β signaling.

To explore the contribution of Lys171 in β-arrestin 2 to the regulation of cGAS, we detected the interaction between β-arrestin 2 mutants and cGAS and found that the interaction was enhanced by K171R mutant (Fig. 5h). Meanwhile, we observed that the DNA-binding ability of cGAS was further enhanced by

K171R mutant but abrogated by K171Q mutant (Fig. 5i), as well as the dimerization of cGAS (Fig. 5j), the production of cGAMP (Fig. 5k), and the subsequent dimerization of STING (Fig. 5l), indicating that the deacetylation of β-arrestin 2 at Lys171 residue was necessary for promoting cGAS activation. Meanwhile, K171R mutant further enhanced the interaction between TBK1 and TRAF3 compared to WT β-arrestin 2 (Supplementary Fig. 3g). Next, we analyzed the acetylation of β-arrestin 2 during virus infection and found that β-arrestin 2 was rapidly deacetylated after infection (Fig. 5m and Supplementary Fig. 3h). We also found that ubiquitination of β-arrestin 2 was significantly increased 12 h after infection, which was coincident with the degradation of β-arrestin 2 (Figs. 1a and 5m and Supplementary Fig. 3h). Consistently, we found that the degradation of β-arrestin 2 could be rescued by MG-132 after virus infection in peritoneal macrophages (Supplementary Fig. 3i, j). Furthermore, the acetylation of K171R mutant was significantly decreased and not further reduced after the infection, whereas the acetylation of K171Q mutant remained invariable at a high level after the infection (Fig. 5n). Consistent with the results that the ubiquitination of both K171R and K171Q mutants was abolished, these two mutants were not degraded after the infection (Fig. 5o and Supplementary Fig. 3k), which indicated that the ubiquitination at Lys171 residue induced the degradation of β-arrestin 2. As the deacetylation (1 h after infection) of β-arrestin 2 occurred before the ubiquitination and degradation (12 h after infection) during infection, we supposed that Lys171 was deacetylated before being ubiquitinated to positively regulate the IFN-β signaling.

**Carvedilol promoted the antiviral immunity through β-arrestin 2.** It has been reported that carvedilol could increase β-arrestin 2 expression in a rat acute myocardial infarction model and regulate $β_1AR$ signaling[42–44]; therefore, we speculated whether carvedilol could play a role in virus infection through β-arrestin 2. We first confirmed the function of carvedilol against virus infection in vivo. The results demonstrated that carvedilol promoted the production of *Ifnb*, *Isg15*, and *Ccl5* and reduced the virus titers in the spleens, livers, and lungs of WT mice after virus infection (Fig. 6a–d). Consistent with this finding, the injury in the lungs (Fig. 6e) and spleens (Fig. 6f) was less, and the survival rates were higher (Fig. 6g, h) after treatment with carvedilol. In agreement with the in vivo results, the production levels of *Ifnb*, *Isg15*, and *Ccl5* were much higher in the carvedilol-treated WT peritoneal macrophages (Fig. 7a, b), whereas the virus titers were lower (Fig. 7c, d) after the infection. Furthermore, carvedilol increased the phosphorylation of TBK1, IRF3, IRF7, and STING (Fig. 7e, f) and the dimerization and translocation of IRF3 (Fig. 7g–j) after the infection. We also detected the production of *Ifnb*, *Isg15*, and *Ccl5* in spleens, livers and lungs in $Arrb2^{-/-}$ mice. We found that carvedilol had no effect on the production of above cytokines in $Arrb2^{-/-}$ mice after HSV-1 infection

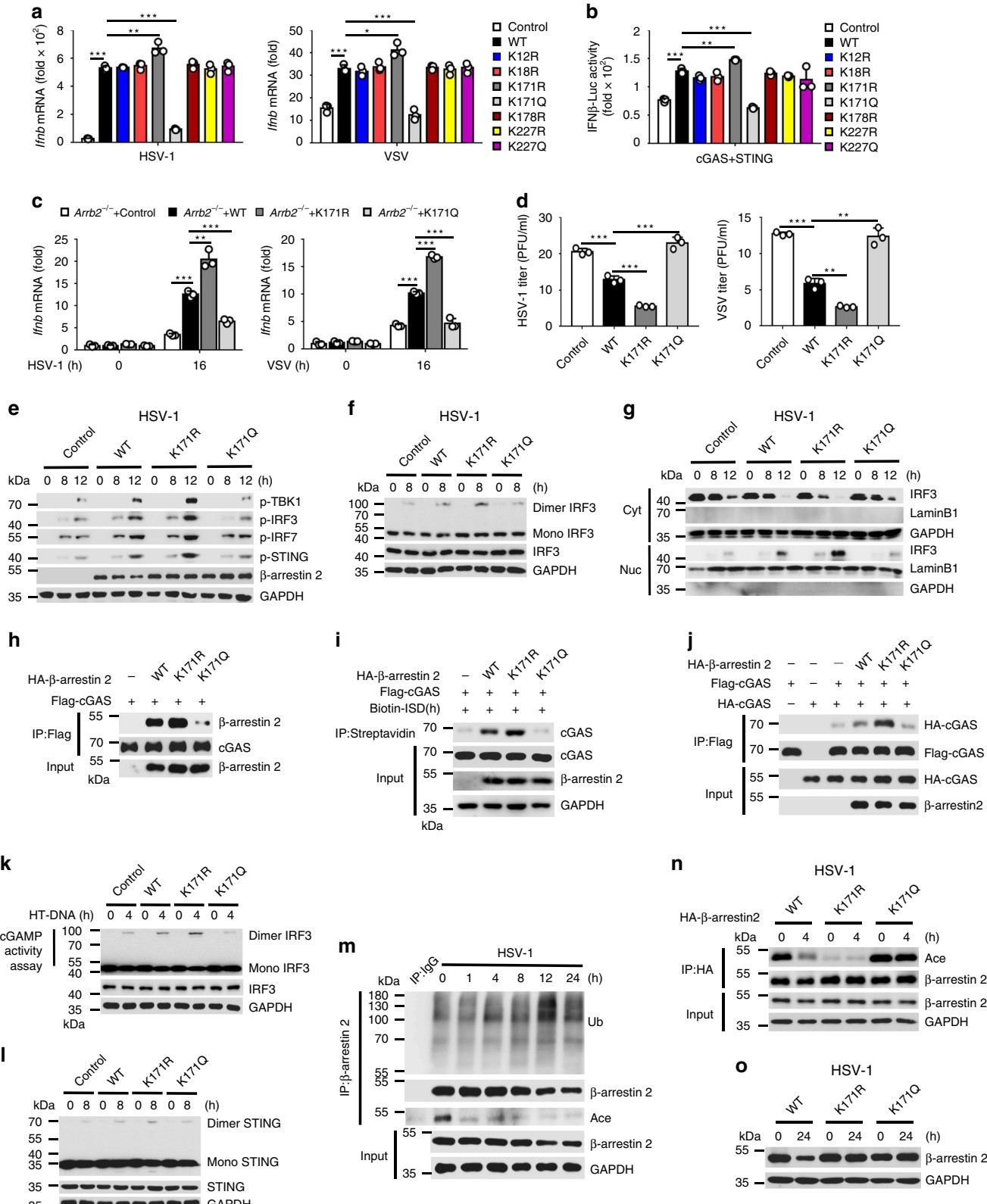

As β-arrestin 2 expression was higher in the carvedilol-treated group than in the control group 12 h after the infection (Fig. 7e, f), we hypothesized that β-arrestin 2 degraded by virus infection was rescued by carvedilol to positively regulate the IFN-β signaling and the antiviral innate immune response. Furthermore,

**Fig. 5 Deacetylation of β-arrestin 2 at Lys171 enhanced the antiviral immune response. a** *Ifnb* mRNA level in HeLa cells transfected with control vector, β-arrestin 2, or β-arrestin 2 mutants and infected with HSV-1 (left) or VSV (right) for 16 h (***$P < 0.0001$, **$P = 0.0067$, ***$P < 0.0001$, left panel; ***$P = 0.0003$, *$P = 0.0133$, ***$P = 0.0003$, right panel). **b** Luciferase assay of IFN-β activation in HEK293T cells expressing various vectors (***$P < 0.0001$, **$P = 0.0017$, ***$P < 0.0001$). **c** *Ifnb* mRNA level in *Arrb2$^{-/-}$* MEFs transfected with control vector, β-arrestin 2, or β-arrestin 2 mutants and infected with HSV-1 (left) or VSV (right) for 16 h (***$P < 0.0001$, **$P = 0.0032$, ***$P = 0.0003$, left panel; ***$P < 0.0001$, $< 0.0001$, $= 0.0005$ in sequence, right panel). **d** The virus titers as in **c** for 72 h (***$P = 0.0006$, $= 0.0003$, $= 0.0007$ in sequence, left panel; ***$P < 0.0001$, *$P = 0.0014$, 0.0012 in sequence, right panel). **e** Immunoassay of lysates of *Arrb2$^{-/-}$* MEFs transfected as in **c** and infected with HSV-1 for indicated times. **f** Immunoblot analysis of monomeric and dimeric IRF3 as in **e**. **g** Immunoblot analysis of nuclear and cytoplasmic fractions as in **e**. **h** Immunoassay of lysates of HEK293T cells expressing various vectors. **i** Immunoassay of cell lysates and streptavidin-precipitated proteins from HEK293T cells transfected with various vectors and stimulated with biotin-ISD. **j** Immunoassay of lysates of HEK293T cells expressing various vectors. **k** cGAMP activity measured by extracts from HT-DNA-stimulated (6 h) *Arrb2$^{-/-}$* MEFs transfected with various vectors. **l** Immunoblot analysis of monomeric and dimeric IRF3 as in **f**. **m** Immunoassay of lysates of peritoneal macrophages infected with HSV-1 for indicated times. **n** Immunoassay of lysates of *Arrb2$^{-/-}$* MEFs transfected as in **l** and infected with HSV-1 for 4 h. **o** Immunoassay of lysates as in **n** for 24 h. Data are representative of at least three independent experiments (mean ± SEM in **b**–**d**, $n = 3$). Two-tailed unpaired Student's *t*-test.

we evaluated the molecular mechanism of carvedilol on the β-arrestin 2-cGAS–STING–IFR3 axis and found that carvedilol significantly enhanced the interaction between β-arrestin 2 and cGAS (Fig. 7k), the DNA-binding ability of cGAS (Fig. 7l), the production of cGAMP (Fig. 7m), and the dimerization of STING (Fig. 7n). In consideration of the pivotal role of Lys171, we also detected the deacetylation of β-arrestin 2 after carvedilol treatment and observed that carvedilol promoted the deacetylation of β-arrestin 2 during infection (Fig. 7o, p).

To elucidate the relationship between carvedilol and Lys171 of β-arrestin 2, we infected *Arrb2$^{-/-}$* MEFs with HSV-1 and VSV, which were overexpressed with WT β-arrestin 2 and its mutants, and treated with carvedilol before lysed. The results showed that carvedilol only enhanced the activity of WT β-arrestin 2 but not β-arrestin 2 mutants in the production of *Ifnb, Isg15, Ccl5*, virus titers, the phosphorylation of TBK1, IRF3, IRF7, STING, the dimerization, and translocation of IRF3 (Supplementary Fig. 5a-g). Furthermore, we detected the interaction between β-arrestin 2 mutants and cGAS, the binding activity between cGAS and dsDNA, the production of cGAMP, and the dimerization of STING, and found that the function of WT β-arrestin 2, but not β-arrestin 2 mutants, was further promoted by carvedilol (Supplementary Fig. 5h-k). Meanwhile, the interaction between TBK1 and TRAF3 was promoted not only by carvedilol alone, but also through β-arrestin 2 (Supplementary Fig. 5l). And also, the interaction between TBK1 and TRAF3 was further promoted by K171R mutant compared to WT β-arrestin 2, while there was no difference with carvedilol treatment (Supplementary Fig. 5m). Together, we thought that carvedilol activated β-arrestin 2 through Lys171. These data revealed that carvedilol positively regulated IFN-β signaling and antiviral immunity by preventing the degradation of β-arrestin 2 and promoting the deacetylation of β-arrestin 2 at Lys171, thus indicating that β-arrestin 2 could be a novel drug candidate for the treatment of virus infection.

## Discussion

Studies have reported that β-arrestin 2 is widely involved in a variety of signaling pathways and functions in multiple diseases, including nervous and metabolic disorders and several cancers[34,46,47]. Furthermore, β-arrestin 2 inhibits the activation of the TRAF6-NF-κB pathway by preventing the self-oligomerization of TRAF6 or inhibiting the release of p65 from IκB[40,41]. However, whether and how β-arrestin 2 regulates the host antiviral innate immune response remain unknown. In the present study, we found that *Arrb2$^{-/-}$* mice exhibited lower IFN-β production, higher virus titers in tissues, and decreased survival rates than their WT littermates. Moreover, β-arrestin 2 promoted the activation of the IFN-β signaling pathway by targeting cGAS,

which was completely different from the function of β-arrestin 2 in the NF-κB signaling pathway. From the perspective of viruses, they were so smart that they could inhibit the expression of β-arrestin 2 by regulating its acetylation and ubiquitination to evade the host immune response. In addition, except that viruses inhibited positive regulators, such as β-arrestin 2, to achieve immune evasion in the present study, some other viruses also use the opposite strategy that enhances the expression of negative regulators to eventually evade the host immune response. For example, one of our previous studies demonstrated that virus-induced PLCβ2 negatively regulates virus-induced proinflammatory responses by inhibiting the activation of TAK1[48]. However, while the devil climbs a post, the priest climbs ten. The clinically proven drug carvedilol rescued the degradation of β-arrestin 2 and promoted the activation of IFN-β signaling and the antiviral immune response, indicating that it might be an efficient antiviral drug candidate to treat virus infection.

We detected the activation of three major signaling pathways in virus infection, including the cGAS–STING, the NF-κB, and MAPK pathways, and found that β-arrestin 2 promoted only the activation of the cGAS–STING pathway but had no effect on the other two pathways. In addition, we observed that β-arrestin 2 enhanced the activation of the cGAS–STING pathway by promoting the recognition of dsDNA by cGAS and the subsequent production of cGAMP. However, the detailed mechanism through which β-arrestin 2 regulates the activation of cGAS and the pivotal residues in cGAS that mediates the activation still need to be further elucidated. Moreover, no study has yet confirmed the enzymatic activity of β-arrestin 2, and it is necessary to further elucidate whether β-arrestin 2 directly regulates or recruits other enzymes to regulate cGAS activation.

We observed that β-arrestin 2 was ubiquitinated and degraded 12 h after infection, whereas it was deacetylated only 1 h after infection, indicating that β-arrestin 2 could normally promote the activation of IFN-β signaling and the antiviral immune response at the early stage of infection, which was decreased by virus-induced ubiquitination and degradation of β-arrestin 2 in the late stage. However, this type of degradation was not a thorough process. In this case, β-arrestin 2 was still expressed at a relatively lower level so that it could promote the antiviral immune response to some extent. We suppose that this smart and accurate strategy of immune evasion was developed during the co-evolution of host and viruses to achieve their symbiosis. This mechanism also preserved the antiviral immune response at a relatively lower level but avoided the production of a strong cytokine storm, which was another smart strategy for the viruses to achieve the symbiosis with the host. Based on the results of mass spectrometry analysis, we assumed that Lys171 is an important residue as it could be acetylated and ubiquitinated,

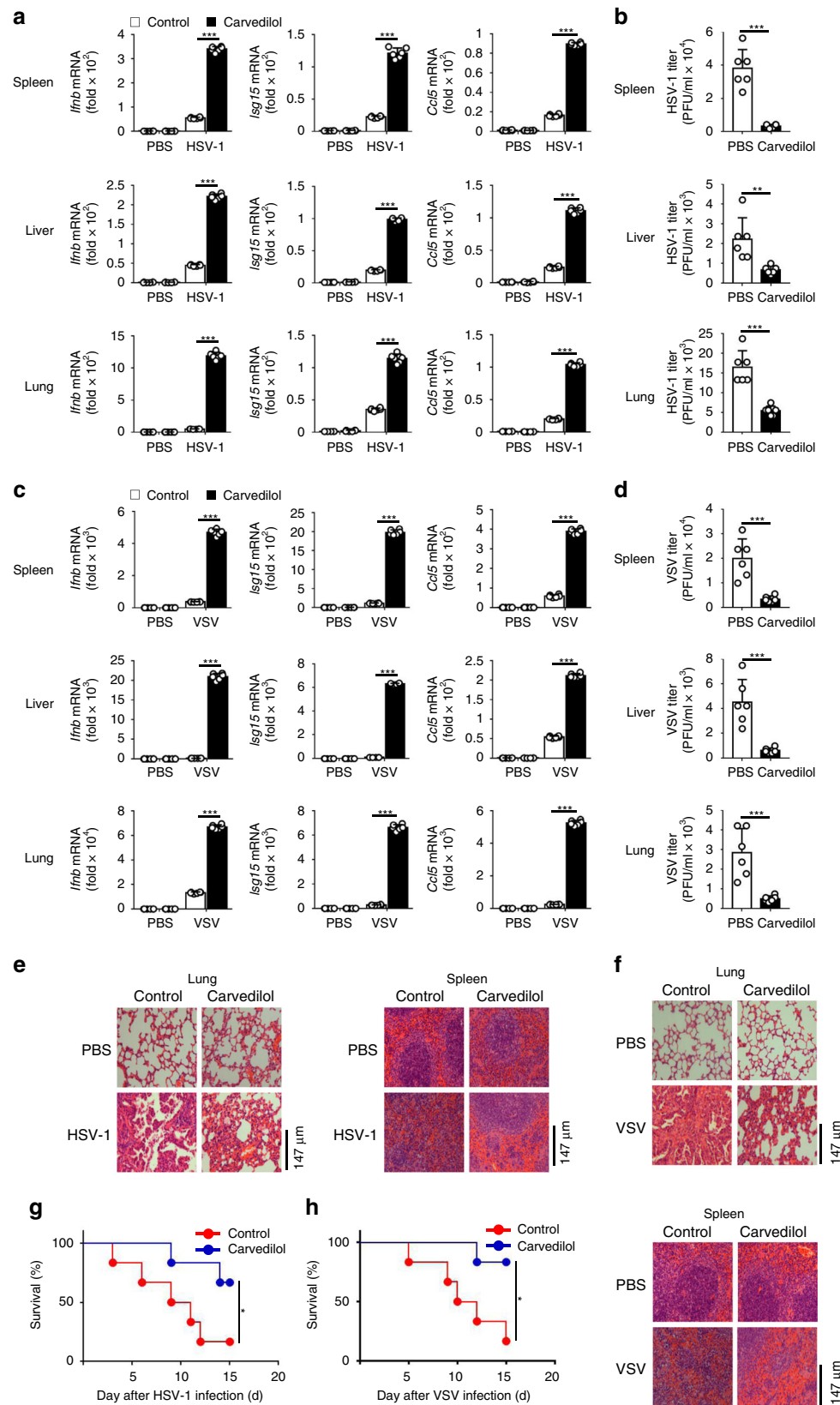

which mediated the acetylation and degradation of β-arrestin 2. We found that β-arrestin 2 was rapidly deacetylated after virus infection. The acetylation-defective mutant of β-arrestin 2 (K171R) enhanced but the acetylation-mimetic mutant (K171Q) impaired the activation of IFN-β signaling compared with the WT β-arrestin 2. We also found that K171R and K171Q were not degraded after virus infection. Thus, the acetylation and ubiquitination of β-arrestin 2 at Lys171 might mediate the function of

**Fig. 6 Carvedilol promoted the antiviral immunity in vivo. a** *Ifnb*, *Isg15*, and *Ccl5* mRNA levels in the spleens, livers, and lungs of WT 6-week-old male mice ($n = 6$ per group) intraperitoneally injected with PBS or HSV-1 ($2 \times 10^8$ PFU per mouse) for 24 h and then treated with PBS or carvedilol for another 24 h (***$P < 0.0001$ in all panels). **b** The virus titers as in **a** (***$P < 0.0001$, up panel; **$P = 0.0059$, middle panel; ***$P < 0.0001$, below panel). **c** VSV ($5 \times 10^8$ PFU per mouse) infection as in **a** (***$P < 0.0001$ in all panels). **d** The virus titers as in **c** (***$P = 0.0005$, up panel; ***$P = 0.0004$, middle panel; ***$P = 0.0009$, below panel). **e**, **f** Microscopy of hematoxylin-and eosin-stained lung and spleen sections as in **a** or **c**. **g**, **h** Survival of WT 6-week-old male mice ($n = 6$ per group) infected intraperitoneally with a high dose of HSV-1 ($5 \times 10^8$ PFU per mouse) **g** (*$P = 0.0485$) or VSV ($8 \times 10^8$ PFU per mouse) (**h**) (*$P = 0.0196$) and treated with PBS or carvedilol 24 h after infection and monitored for 15 d. Data are representative of at least three independent experiments (mean ± SEM, two-tailed unpaired Student's *t*-test in **a**–**d** or Kaplan–Meier analysis in **g**, **h**).

β-arrestin 2 in the activation of the IFN-β signaling pathway, the specific mechanism of which needs to be further investigated.

In addition, we demonstrated that the β-blocker carvedilol rescued the expression of the virus infection-degraded β-arrestin 2, which further enhanced the antiviral immune response. Carvedilol also promoted the deacetylation of β-arrestin 2 after virus infection, which might partially explain why carvedilol could increase the expression of β-arrestin 2 and improve the antiviral immune response. These findings revealed that carvedilol might promote the antiviral innate immune response by enhancing β-arrestin 2 expression, which may be an outstanding example of the novel use of a conventional drug to treat viral infectious diseases.

Altogether, our study findings have identified β-arrestin 2 as a positive regulator of precise control of the cellular antiviral immune response through regulating cGAS activation and IFN-β production. This result should open up new perspectives on the regulation of the cGAS–STING signaling pathway. Furthermore, our study has shed light on the novel mechanism of immune evasion by viruses and the possibility for the novel use of the conventional drug carvedilol as a potential drug candidate for the treatment of viral infectious diseases.

## Methods

**Reagents and plasmids**. The following chemical reagents and antibodies were used in this study: anti-TBK1 (3504), anti-phospho-TBK1 (5483), anti-phospho-IRF3 (29047), anti-phospho-IRF7 (24129), anti-phospho-p65 (3033), anti-phospho-p38 (9215), anti-phospho-Erk1/2 (9101), anti-STING (13647), anti-cGAS (15102), anti-RIG-I (4200), anti-phospho-STING (72971), and anti-β-arrestin 2 (3857 S; all from Cell Signaling Technology); anti-β-arrestin 2 (ab54790; from Abcam) are diluted in the ratio of 1:1000; anti-Flag M2 Affinity Gel (A2220), anti-HA (H9658), anti-HA (H6908), and anti-Flag (F7425; all from Sigma-Aldrich) are diluted in the ratio of 1:10000; anti-GST (CW0085M) and anti-His (CW0083M; both from Cwbio) are diluted in the ratio of 1:10,000; and Protein G Sepharose 4 Fast Flow (GE Healthcare). Expression constructs for Flag-RIG-I, Flag-RIG-I-N, Flag-TBK1, Flag-IRF3, Flag-cGAS, Flag-TRAF3, HA-cGAS, HA-CK1ε, and HA-Ubs were obtained from Dr. B. Ge (Tongji University, Shanghai, China), and Flag-STING was obtained from Dr. B. Sun (Shanghai Institute of Biochemistry and Cell Biology, Shanghai, China). Site-directed point mutagenesis of β-arrestin 2 was performed using the KOD Plus Mutagenesis Kit (SMK101, TOYOBO) according to the manufacturer's instructions.

**Cells and viruses**. Mouse peritoneal macrophages, MEFs, and RAW264.7, L929, and HEK293T cells were maintained in Dulbecco's Modified Eagle's Medium (DMEM; Hyclone) supplemented with 10% (v/v) heat-inactivated FBS (Gibco) and 100 U/ml penicillin and streptomycin (Hyclone). HeLa cells were cultured in RPMI-1640 medium supplemented with 10% (v/v) heat-inactivated FBS (Gibco) and 100 U/ml penicillin and streptomycin (Hyclone). Vero cells were cultured in DMEM supplemented with 3% (v/v) heat-inactivated FBS (Gibco). HSV-1, VSV, and SeV were obtained from Dr. B. Ge (Tongji University, Shanghai, China).

**Mouse strains**. Homozygous *Arrb2*$^{-/-}$ mice (from Dr. G. Pei, Tongji University, Shanghai, China) were bred under specific pathogen-free conditions at the Shanghai Research Center for Biomodel Organisms. Six-week-old male specific pathogen-free (SPF) *Arrb2*$^{-/-}$ mice and their WT littermates used in the experiments were bred separately and put to death by cervical vertebra dislocation. All animal studies were approved by the Institutional Animal Care and Use Committee of Fudan University.

**Isolation of mouse peritoneal macrophages and MEFs**. For preparing mouse peritoneal macrophages, 6-week-old homozygous *Arrb2*$^{-/-}$ mice and their WT littermates were injected with 1.5 ml Brucella broth (4%) intraperitoneally. After 3 days, the peritoneal lavage fluid was collected from the mice and washed three times with PBS. Peritoneal macrophages were cultured in DMEM supplemented with 10% FBS. For the preparation of MEFs, 13-day-old embryos of *Arrb2*$^{-/-}$ mice or their WT littermates were digested with 0.25% trypsin at 4 °C overnight. After percussion blending, MEFs were washed three times with PBS and then cultured in DMEM supplemented with 10% (v/v) heat-inactivated FBS (Gibco) and 100 U/ml penicillin and streptomycin (Hyclone).

**Virus infection**. For in vitro virus infection, L929, HeLa, and RAW264.7 cells and peritoneal macrophages ($2 \times 10^6$ cells) were cultured in DMEM for 12 h and infected with HSV-1 (MOI = 10) or VSV (MOI = 1) or SeV (100 hemagglutination units [HAU]/ml) for the indicated times. For in vivo virus infection, 6-week-old homozygous *Arrb2*$^{-/-}$ mice and their WT littermates were injected with HSV-1 ($2 \times 10^7$ PFU/mouse) and VSV ($5 \times 10^8$ PFU/mouse) for the indicated times.

**Transfection**. HEK293T cells were transiently transfected with polyethylenimine (PEI; 23966-2; Polysciences) according to the manufacturer's instructions. HeLa or L929 cells were transfected with Lipofectamine 2000 (11668; Invitrogen), and RAW264.7 and MEFs were transfected with Lipofectamine 3000 (L3000; Invitrogen). Double-stranded oligonucleotides corresponding to the target sequence (*Arrb2*, GGAACUCUGUGCGGCUUAUTT) were cloned into the plasmid pLKO.1.

**RNA-seq library preparation, sequencing, and data processing**. WT and *Arrb2*$^{-/-}$ PMs treated for 16 h with HSV-1 before they were lysed with RNAiso Plus according to the manufacturers' instructions. The lysates were sent to Personalbio for cDNA library construction and sequencing by Illumina NovaSeq 6000. RNA-sequencing reads were first trimmed to remove poly(A) and unqualified reads with *cutadapt*, then aligned to Mus musculus GRCm38 genome with *Ensembl*. The numbers of counts were summarized at the gene level using *HTseq*. Gene expression values were computed from fragments per kilo bases per million fragments (FPKM) values produced by addition of a pseudocount of 1 and log2 transformation of the results. These FPKM values were used for drawing the heatmap with the *pheatmap* R package. Paired differential gene expression analyses were performed with *DEseq* R package by addition of fold change > 2 and *p*-value < 0.05. Volcano plots of these differential genes were drawing with *ggplots2* R package. GO or KEGG enrichment analyses of differential genes were performed using *topGO* or *KAAS*, respectively. The terms or pathways involved in host antiviral immune response with fold change > 2 and *p*-value < 0.05 were used to drawing GO term enrichment or KEGG pathway enrichment. The PPI analysis of differential genes was performed with STRING database and PPI network model was drawn with *Cytoscape*.

**Quantitative real-time PCR**. Cells were incubated for 12 h with DMEM and then infected with the viruses for the indicated times. Total RNA was isolated using RNAiso Plus (9109; Takara) according to the manufacturer's instructions. Then, 1 μg of RNA was reverse-transcribed using the PrimeScript™ RT Reagent Kit (RR037; Takara) to generate cDNA. A LightCycler (LC480; Roche) and a SYBR RT-PCR Kit (QPK-212; Toyobo) were used for quantitative real-time RT-PCR analysis. Gene amplification was performed using the ΔΔCt method. Gene expression was normalized to that of GAPDH. The primer sequences used to amplify human and mouse genes are described in Supplementary Table 1.

**Immunoprecipitation and western blotting**. Cells were transfected with the indicated plasmids. After 48 h, cells were lysed in a lysis buffer (50 mM Tris (pH 7.4), 150 mM NaCl, 1% Triton X-100, and 1 mM EDTA (pH 8.0)) supplemented with a protease cocktail (Roche, 04693159001) consisting of 1 mM PMSF, 1 mM Na$_3$VO$_4$, and 1 mM NaF for 30 min on ice. The lysates were centrifuged at 13,200 rpm for 15 min at 4 °C to remove the debris. Cell lysates were incubated with anti-Flag M2 Affinity Gel or Protein G Sepharose 4 Fast Flow plus pre-specified antibodies at 4 °C overnight. For immunoprecipitation of endogenous protein, mouse peritoneal macrophages or HeLa cells were lysed and incubated

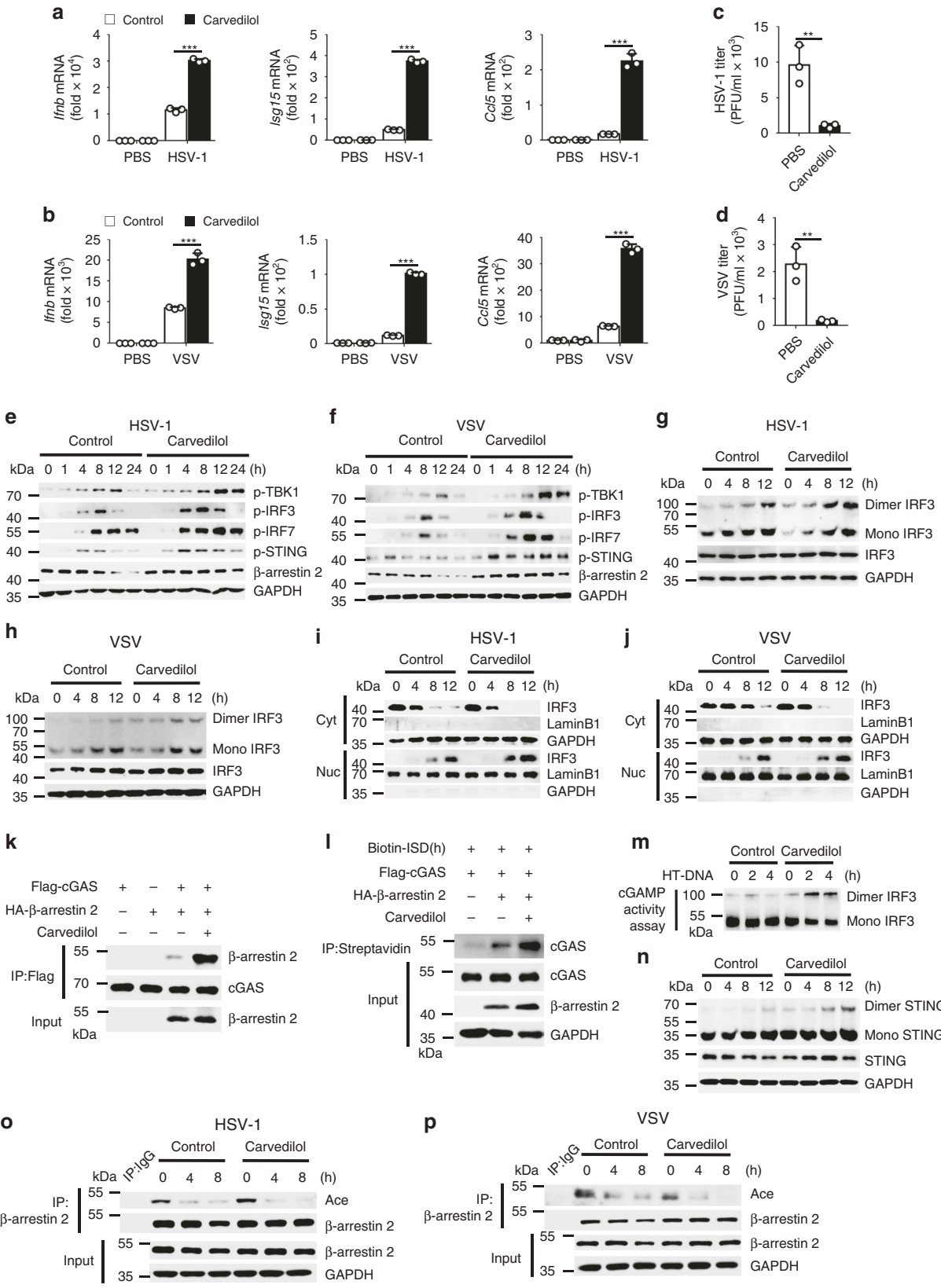

with the specified antibodies and Protein G Sepharose 4 Fast Flow at 4 °C overnight. The sepharose samples were centrifuged and washed three times with ice-cold PBST buffer (1% Triton X-100 in PBS). Precipitates or cell lysates were boiled in 1× SDS loading buffer at 100 °C for 10 min and then analyzed by immunoblotting.

**GST pull-down.** His or GST fusion proteins were expressed in BL-21(DE3) (Tiangen Biotech) according to the manufacturer's instructions. GST fusion proteins were lysed by ultrasonication, incubated with GST beads for 4 h at 4 °C, and then washed three times with TBST to collect the precipitates. Purified His fusion protein or lysates of mouse peritoneal macrophages were incubated with the

**Fig. 7 Carvedilol promoted cGAS–STING activation through β-arrestin 2. a, b** *Ifnb*, *ISG15*, and *Ccl5* mRNA in WT peritoneal macrophages infected for 16 h with HSV-1 **a** (***$P < 0.0001$ in all panels) or VSV **b** (***$P = 0.0002$, ***$P < 0.0001$, ***$P < 0.0001$) and then treated with PBS or carvedilol. **c, d** The virus titers as in **a** or **b** for 72 h (**$P = 0.0058$ in **c**, **$P = 0.0050$ in **d**). **e, f** Immunoblot of lysates of peritoneal macrophages from WT mice infected with HSV-1 **e** or VSV **f** for indicated times and treated with PBS or carvedilol. **g, h** Immunoblot analysis of monomeric and dimeric IRF3 as in **e** or **f**. **i, j** Immunoblot analysis of nuclear and cytoplasmic fractions as in **e** or **f**. **k** Immunoassay of lysates of HEK293T cells transfected with various vectors and treated with PBS or carvedilol. **l** Immunoassay of cell lysates and streptavidin-precipitated proteins from HEK293T cells transfected with various vectors and stimulated with biotin-ISD and carvedilol. **m** cGAMP activity measured by extracts from HT-DNA-stimulated (6 h) WT peritoneal macrophages treated with PBS or carvedilol. **n** Immunoblot analysis of monomeric and dimeric STING as in **g**. **o, p** Immunoassay of lysates of peritoneal macrophages infected with HSV-1 **o** or VSV **p** for indicated times and treated with PBS or carvedilol. Data are representative of at least three independent experiments (mean ± SEM in **a–d**, $n = 3$). **$P < 0.01$ and ***$P < 0.001$, two-tailed unpaired Student's $t$-test.

collected precipitates for 4 h at 4 °C. After centrifugation and washing, the beads were boiled in 1× SDS loading buffer at 100 °C for 10 min and then analyzed by immunoblotting.

**Native PAGE.** The dimerization assay was performed as described. In brief, mouse peritoneal macrophages or L929 cells were cultured in DMEM for 12 h and infected with virus for the indicated times. Cells were harvested using 100 µl ice-cold lysis buffer (50 mM Tris, pH 7.4; 150 mM NaCl; 1% Triton X-100; 1 mM EDTA, pH 8.0; and a protease inhibitor cocktail consisting of 1 mM PMSF, 1 mM $Na_3VO_4$, and 1 mM NaF). After centrifugation at $13,000 × g$ for 15 min at 4 °C, the supernatants were quantified and diluted with 2× native PAGE sample buffer (125 mM Tris-HCl, pH 6.8; 30% glycerol; and 0.1% bromophenol blue). Then, 30 µg of protein was applied to a pre-run 7.5% native gel. After electrophoresis, the proteins were transferred onto a nitrocellulose membrane for immunoblotting.

**In vitro assay for cGAMP activity.** WT or *Arrb2*$^{-/-}$ mouse peritoneal macrophages were left untreated or treated with HT-DNA for 6 h and homogenized by douncing in the lysis buffer (10 mM Tris-HCl, pH 7.5, 10 mM KCl, 1.5 mM $MgCl_2$, 1 mM DTT, and 1 mM PMSF) at 4 °C. The homogenates were centrifuged at 10,000 rpm for 20 min at 4 °C. After heating at 95 °C for 5 min, the supernatant was centrifuged again at 12,000 rpm for 10 min to remove precipitants. The lysates were mixed with fresh WT mouse peritoneal macrophages in 12.5 µl reaction buffer containing 2 mM ATP, 1 U/µl Benzonase, and 2 ng/µl PFO for 2 or 4 h at 37 °C. The cells were then subjected to the Native PAGE assay.

**Cell staining and confocal microscopy.** HeLa cells were transfected with the appropriate plasmids for 48 h and infected with HIV-1 or VSV for the indicated times. Peritoneal macrophages were infected with viruses directly. Cells were fixed with 4% formaldehyde for 20 min at room temperature, permeabilized for 30 min in PBS containing 0.3% Triton X-100, and then blocked for 1 h at 4 °C in a blocking buffer (1% BSA in PBS). Then, the cells were incubated with the indicated antibodies at 4 °C overnight and secondary antibodies at room temperature for 1 h. After staining with DAPI, images were obtained using a Leica TCS SP5 confocal laser microscopy system[35].

**Dual-luciferase reporter assay.** HEK293T cells were transiently transfected with pRL-IFN-β–Luc or pRL-ISRE–Luc, pRL-TK, and the indicated plasmids for 24 h. Dual-luciferase reporter assay system (RG028; Beyotime) was used to detect the luciferase activity according to the manufacturer's instructions.

**Electrophoretic mobility shift assay (EMSA).** To conduct the EMSA, we firstly prepared the lysates of competent *E. coli* transformed with β-arrestin 2 and cGAS plasmids for the DNA–protein conjugation reaction and measured the protein concentration by BCA protein assay kit (Beyotime, P0012).We synthesized biotin-labeled ISD probe for EMSA assay (Sangon Biotec). Then we executed the DNA–protein binding reactions using EMSA Gel-Shift Kit (Beyotime,GS009) following the manufacturer's instructions at 25 °C. Finally, the reaction mixture was separated through 4% non-denaturing PAGE gels and developed.

**Statistical analysis.** Data are expressed as mean ± standard error of the mean (SEM). Prism 6 (GraphPad) is used for statistical analyses. The statistical tests conducted in this study are indicated in the figure legends as follows: *$P < 0.05$; **$P < 0.01$; ***$P < 0.001$, two-tailed unpaired Student's $t$-test. Leica TCS SP8 confocal laser microscopy system is used for Immunohistochemistry data. ViiA 7 Software v1.2.4 on ViiA 7DX is used for qRT-PCR data. ImageQuant LAS 4000mini and Amersham Imager 600 is used for western blot data. RNA-seq data are analyzed by cutadapt, Ensembl, HTseq, pheatmap R package, DEseq R, ggplots2 R, topGO, KAAS, STRING database, and Cytoscape as described in RNA-seq library preparation, sequencing and data processing section. Illumina NovaSeq 6000 is used for RNA sequencing.

**Reporting Summary.** Further information on research design is available in the Nature Research Reporting Summary linked to this article.

## Data availability
The authors declare that the data supporting the findings of this study are available within the paper and its supplementary information files. Source data are provided with this paper. All other data that support the results of the study will be available from the corresponding author upon reasonable request. The RNA-seq data have been deposited in Gene Expression Omnibus (GEO) under the GSE159735. Source data are provided with this paper.

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

## Acknowledgments

We thank Dr. G. Pei (Tongji University, Shanghai, China) for providing *Arrb2⁻/⁻* mice, Dr. B. Ge (Tongji University, Shanghai, China) for providing SeV, VSV, HSV-1, and expression constructs for Flag-RIG-I, Flag-N-RIG-I, Flag-TBK1, Flag-IRF3, Flag-cGAS, Flag-TRAF3, HA-cGAS, HA-CK1ε, and HA-ubiquitinations and Dr. B. Sun for providing Flag-STING from (Shanghai Institute of Biochemistry and Cell Biology, Shanghai, China). This work was supported by grants from the National Natural Science Foundation of China (project 31972900 and 31670901), the Program for Professor of Special Appointment (Eastern Scholar) at Shanghai Institutions of Higher Learning (program TP2016007), the Outstanding Youth Training Program of Shanghai Municipal Commission of Health and Family Planning (program 2017YQ012), Innovative Research Team of High-level Local Universities in Shanghai, the National Key Research and Development Program of China (program 2016YFC1305103 and 2018YFC1705505), and the National megaproject on key infectious diseases (program 2017ZX10202102).

## Author contributions

Y.Z. and D.Y. designed this study; Y.Z. and M.L. performed the experiments assisted by L.L., G.Q., Y.W., Z.C., J.L., C.F., and F.H.; D.G., Q.Z., and Y.C. contributed to discussions and agreement with the conclusions; Y.Z. and D.Y. analyzed the data and wrote the manuscript; and all authors discussed the results and commented on the manuscript.

## Competing interests

The authors declare no competing interests.
