## [Peer Review File · Nature Communications]

REVIEWER COMMENTS

Reviewer #1 (STING/cGAS and RIG-I signaling) (Remarks to the Author):

Zhang et al report analysis of the role of beta-arrestin in activating innate immune signaling. They provide data showing that loss of beta-arrestin results in impaired activation of the IFN-beta promoter in response to different immune stimuli, whereas overexpression leads to an increase of activation. A particular strong differential effect is seen with respect to ISD, a cGAS-STING activating DNA ligand, and poly-IC, which activates RIG-I or MDA5 as a function of its length. The analysis includes cell lines but also an in vivo mouse model. Overall, this is an important area of research and the authors have compiled a large and comprehensive set of experimental data. Before I can recommend publication, the following points should be addressed:

Experimental:

1) Fig. 2C,D: Did the authors add a control with transfection of a non stimulating DNA, or empty transfection reagent? I wonder why poly-IC is as efficient as ISD, given that beta-arrestin interacts with cGAS and not other components?

2) Fig. 4G: How can the authors rule out that the observed increase in luciferase activity is not due to cGAS being activated by increased amounts of transfected DNA, rather than by beta-arrestin protein? I.E. by assuring that the total amount of transfected DNA at a given time is constant?

3) A main point is the interaction of beta-arrestin with cGAS and enhancement of cGAS' DNA binding activity. I think the data should be strengthened with a demonstration that beta-arrestin enhances cGAS' DNA binding activity in vitro using purified proteins, for instance by EMSAs. This should be relatively straightforward since the authors have purified components and would rule out indirect effects that could take place in pull downs from cell extracts.

4) What is explanation that poly-IC triggered innate immune response is regulated by beta-arrestin? Can the authors show that this requires cGAS as well, or does it involve another pathway?

Editorial:

- abstract: I am not sure if "smart" is a good term to describe viruses, perhaps consider rephrasing
- Introduction is a bit redundant (lines 60 and 67)
- line 70: cGAS has only one MB21 domain, and at least human cGAS has three DNA binding sites.
- line 79: "location is not constant". This is a bit confusing as all molecules in a cell somehow move around. Consider reformulating.

Reviewer #2 (STING/cGAS, innate immunity) (Remarks to the Author):

In the manuscript, Yihua Zhang et al reported that carvedilol improved the antiviral innate immune response through β -arrestin 2-cGAS-STING-IRF3 axis. The authors began their study by the finding that deficiency of β -arrestin 2 diminished the antiviral immune response both in vivo and in vitro, while exogenous overexpression of β -arrestin 2 positively regulated IFN- β signaling. Mechanistically, the authors found that carvedilol rescued β -arrestin 2 expression and accelerated its deacetylation at Lys171, which promoted the DNA-binding of cGAS, the production of cGAMP, the dimerization of STING, as well as the dimerization and nuclear translocation of IRF3. Overall, the conclusions were of interest to the field of innate immunity. However, the mechanism that β -arrestin 2 targeted cGAS-

cGAMP-STING (DNA sensing) was not reasonable to explain its role in supporting antiviral immune response induced by RNA viruses or poly (I:C), therefore the effects of some other mechanisms should be included. Besides, it was suggested that carvedilol positively regulated IFN- β signaling through β -arrestin 2, but more importantly, further experiments were required to show whether carvedilol could promote cGAS-STING activation in Arrb2 $^{-/-}$ mice. A few major issues needed to be addressed before the manuscript was ready for publication, as described below.

1. Figure 2i, lanes 5-8, the dimer STING increased, and the mono STING unchanged, while the total STING decreased (especially 4 h and 8 h post HSV-1 infection). The authors should explain this paradox or repeat this experiment.
2. Figure 4m (lanes 3 and 4), 4n (lane 8), 5f (lanes 4, 5 and 6), 5k (lanes 4 and 6), 5l (lanes 4 and 6) and s2f (lanes 2, 4 and 6), the changes of dimer IRF3 or STING were not significant, especially noting that the increase of dimer IRF3 or STING was always accompanied by synchronously increased mono STING or IRF3 in some perverse way. In theory, the trend of the changes of the dimer protein should be opposite to that of the mono protein under the constant quantity of total protein (like figure 7m, lanes 5 and 6). So, a repeat of these experiments with the same loadings of mono STING or IRF3 was necessary to make sure that the enhanced dimerization was not due to the increase of the total protein.
3. Figure 4 illustrated the interaction between β -arrestin 2 and cGAS, the authors should continue mapping the interaction sites, both in β -arrestin 2 and cGAS, through co-immunoprecipitation experiments, using different truncations or mutations.
4. Figure 5i, lane 1, the basic interaction between cGAS and ISD was supposed to be detectable (at least stronger than K171Q, lane 4), referring to figure 4i, lane 3. The authors should explain this contradiction.
5. Figure 5m, further evidence was needed to prove the ubiquitination-mediated degradation of β -arrestin 2. For example, perform experiments to test whether the degradation of β -arrestin 2 could be rescued by proteasome inhibitor MG-132.
6. It was necessary to confirm whether carvedilol could still enhance IFN- β signaling and antiviral immunity in β -arrestin 2 deficient cells and mice. In figure 6, the authors should supplement experimental data to demonstrate the loss of effect of Carvedilol on ISGs induction, virus titers and survival rates in Arrb2 $^{-/-}$ mice under the same treatment. Figure 7, same experiments should be performed using Arrb2 $^{-/-}$ +control, Arrb2 $^{-/-}$ +WT, Arrb2 $^{-/-}$ +K171R and Arrb2 $^{-/-}$ +K171Q cells.
7. Figure s3, the duplicate "inhibition of degradation" below in the diagram may be corrected as "promotion of deacetylation" according to the whole text.
8. Check and correct English grammar and usage. For example, line 84, the first "and" was redundant; line 339-342, repetitive description.

Reviewer #1 :

Zhang et al report analysis of the role of beta-arrestin in activating innate immune signaling. They provide data showing that loss of beta-arrestin results in impaired activation of the IFN-beta promoter in response to different immune stimuli, whereas overexpression leads to an increase of activation. A particular strong differential effect is seen with respect to ISD, a cGAS-STING activating DNA ligand, and poly-IC, which activates RIG-I or MDA5 as a function of its length. The analysis includes cell lines but also an in vivo mouse model. Overall, this is an important area of research and the authors have compiled and large and comprehensive set of experimental data. Before I can recommend publication, the following points should be addressed:

We thank the reviewer for careful reading and positive comments about our work.

Question 1: Fig. 2C,D: Did the authors add a control with transfection of a non stimulating DNA, or empty transfection reagent? I wonder why poly-IC is as efficient as ISD, given that beta-arrestin interacts with cGAS and not other components?

Reply 1: We thank the reviewer for pointing this concern. Actually, we added the control vectors and the empty transfection reagent in all of experiments when it was needed. As β -arrestin 2 could promote the activity of luciferase reporters for IFNB or ISRE activation induced by signaling proteins except for IRF3-5D (Fig. 4d,e), we detected the interaction between β -arrestin 2 and these signaling proteins, including RIG-I, MAVS, cGAS, STING, TBK1 and TRAF3, and found that β -arrestin 2 could only bind with cGAS in HEK293T cells(Fig. 4f,g). Meanwhile, GST pull down assay showed that β -arrestin 2 could bind with cGAS directly(Fig. 4h), so we mainly focused on the interaction between cGAS and β -arrestin 2 in our manuscript. However, in the infection assay, β -arrestin 2 could not only bind with cGAS, but also with TBK1 and TRAF3 during virus infection (Extended Data Fig. 2h). Furthermore, we found that the interaction between TBK1 and TRAF3 was promoted by β -arrestin 2 (Extended Data Fig. 2i), which might explain why poly(I:C) was as efficient as ISD. The detailed mechanism involved in this process will be further elucidated in our future work.

Question 2: Fig. 4G: How can the authors rule out that the observed increase in luciferase activity is not due to cGAS being activated by increased amounts of transfected DNA, rather than by beta-arrestin protein? I.E. by assuring that the total amount of transfected DNA at a given time is constant?

Reply 2: We thank the reviewer for pointing this concern. We added the control DNA and the empty transfection reagent in all of experiments when it was needed. We ensured that the total amount of transfected DNA at a given time is constant when we transfected each sample, so that the increased luciferase activity triggered by cGAS plus STING is due to increased amounts of β -arrestin 2.

Question 3: A main point is the interaction of beta-arresting with cGAS and enhancement of cGAS' DNA binding activity. I think the data should be strengthened with a demonstration that Best regards beta-arrestin enhances cGAS' DNA binding activity in vitro using purified proteins, for instance by EMSAs. This should be relatively straightforward since the authors have purified components and would rule out indirect effects that could take place in pull downs from cell extracts.

Reply 3: As per the reviewer's suggestion, we added an EMSA analysis of binding affinity between cGAS and dsDNA. We found that β -arrestin 2 could directly promote the interaction between dsDNA and cGAS (Fig. 4I).

Here, we would like to explain why we did not use EMSA analysis to detect the interaction between cGAS and dsDNA before. We agree with the reviewer that EMSA is very efficient in proving protein-DNA binding activity. But we were worried that cGAS may lose some modification by β -arrestin 2 in vitro, which might influence the increased binding affinity between cGAS and dsDNA that was promoted by β -arrestin 2. On the other hand, boitin-labeled dsDNA is widely used to detected the cGAS-DNA binding activity. So we thought that we chose a proven method and our results and conclusion were credible.

Question 4: What is explanation that poly-IC triggered innate immune response is regulated by beta-arrestin? Can the authors show that this requires cGAS as well, or does it involve another pathway?

Reply 4: As per the reviewer's suggestion, we hypothesized that there might exist some other proteins that could associate with β -arrestin 2. As we explained in repply1, β -arrestin 2 might regulate poly(I:C) triggered innate immune response by interacting with TBK1 and TRAF3 during virus infection. Furthermore, we found that the interaction between TBK1 and TRAF3 was promoted by β -arrestin 2. The detailed mechanism involved in this process will be further elucidated in our future work.

Reviewer #2 :

In the manuscript, Yihua Zhang et al reported that carvedilol improved the antiviral innate immune response through β -arrestin 2-cGAS-STING-IRF3 axis. The authors began their study by the finding that deficiency of β -arrestin 2 diminished the antiviral immune response both in vivo and in vitro, while exogenous overexpression of β -arrestin 2 positively regulated IFN- β signaling. Mechanistically, the authors found that carvedilol rescued β -arrestin 2 expression and accelerated its deacetylation at Lys171, which promoted the DNA-binding of cGAS, the production of cGAMP, the dimerization of STING, as well as the dimerization and nuclear translocation of IRF3. Overall, the conclusions were of interest to the field of innate immunity. However, the mechanism that β -arrestin 2 targeted cGAS-cGAMP-STING (DNA sensing) was not reasonable to explain its role in supporting antiviral immune response induced by RNA viruses or poly (I:C), therefore the effects of some other mechanisms should be included. Besides, it was suggested that carvedilol positively regulated IFN- β signaling through β -arrestin 2, but more importantly, further experiments were required to show whether carvedilol could promote cGAS-STING activation in *Arb2*^{-/-} mice. A few major issues needed to be addressed before the manuscript was ready for publication, as described below.

We thank the reviewer for careful reading and comments about our work.

Question 1: Figure 2i, lanes 5-8, the dimer STING increased, and the mono STING unchanged, while the total STING decreased (especially 4 h and 8 h post HSV-1 infection). The authors should explain this paradox or repeat this experiment.

Reply 1: As per the reviewer's suggestion, we repeat this experiment for several times and found that the dimer STING increased, while the mono and the total STING were unchanged (Fig. 2i).

Question 2: Figure 4m (lanes 3 and 4), 4n (lane 8), 5f (lanes 4, 5 and 6), 5k (lanes 4 and 6), 5l (lanes 4 and 6) and s2f (lanes 2, 4 and 6), the changes of dimer IRF3 or STING were not significant, especially noting that the increase of dimer IRF3 or STING was always accompanied by synchronously increased mono STNG or IRF3 in some perverse way. In theory, the trend of the changes of the dimer protein should be opposite to that of the mono protein under the constant quantity of total protein (like figure 7m, lanes 5 and 6). So, a repeat of these experiments with the same loadings of mono STNG or IRF3 was necessary to make sure that the enhanced dimerization was not due to the increase of the total protein.

Reply 2: We thank the reviewer for careful reading and pointing this concern. As per the reviewer's suggestion, we repeat all these experiment above for several times and found that it was difficult to show the opposite trend of the changes of the dimer protein and the mono protein under the constant quantity of total protein. We could only detect the change of dimer IRF3 or STING, but not the change of mono and total STNG or IRF3.

We agree with the review that "In theory, the trend of the changes of the dimer protein should be opposite to that of the mono protein under the constant quantity of total protein". However, we think it is because that the amount of mono protein is far more than that of dimer protein. It is hard to see the slight change if there are too much mono proteins. Furthermore, in some other studies, the authors also did not see the opposite trend of the dimer protein and the mono protein under the constant quantity of total protein. In these studies, we can only see the change of the dimer protein, the mono protein seems invariable under the constant quantity of total protein¹⁻³. Please see their results below.

Question 3: Figure 4 illustrated the interaction between β -arrestin 2 and cGAS, the authors should continue mapping the interaction sites, both in β -arrestin 2 and cGAS, through co-immunoprecipitation experiments, using different truncations or mutations.

Reply 3: As per the reviewer's suggestion, we constructed a series of truncation mutants of cGAS and β -arrestin 2, respectively. We detected the interaction between full-length β -arrestin 2 and cGAS truncation, the interaction between full-length cGAS and β -arrestin 2 truncation. We found that 1-185 truncation of β -arrestin 2 and 213-522 truncation of cGAS were important for their interaction (Extended Data Fig. 2f, g).

Question 4: Figure 5i, lane 1, the basic interaction between cGAS and ISD was supposed to be detectable (at least stronger than K171Q, lane 4), referring to figure 4i, lane 3. The authors should explain this contradiction.

Reply 4: As per the reviewer's suggestion, we repeat this experiment for several times and found the interaction between cGAS and ISD was detectable (Fig. 5i). We think the reason why we did not detect this is because the interaction is too slight; we need more time to expose the film to see the band.

Question 5: Figure 5m, further evidence was needed to prove the ubiquitination-mediated degradation of β -arrestin 2. For example, perform experiments to test whether the degradation of β -arrestin 2 could be rescued by proteasome inhibitor MG-132.

Reply 5: We fully agree with the reviewer's point. As per the reviewer's suggestion, we detected the expression of β -arrestin 2 in peritoneal macrophages during virus infection and treated with MG132. We found that the degradation of β -arrestin 2 could be rescued by MG-132 after virus infection in

peritoneal macrophages (Extended Data Fig. 3i, j).

Question 6: It was necessary to confirm whether carvedilol could still enhance IFN- β signaling and antiviral immunity in β -arrestin 2 deficient cells and mice. In figure 6, the authors should supplement experimental data to demonstrate the loss of effect of Carvedilol on ISGs induction, virus titers and survival rates in Arrb2^{-/-} mice under the same treatment. Figure 7, same experiments should be performed using Arrb2^{-/-}+control, Arrb2^{-/-}+WT, Arrb2^{-/-}+K171R and Arrb2^{-/-}+K171Q cells.

Reply 6: As per the reviewer's suggestion, we detected the production of *Ifnb*, *Isg15*, *Ccl5* in spleens, livers and lungs in Arrb2^{-/-} mice treated with carvedilol. We found that carvedilol had no effect on the production of above cytokines in Arrb2^{-/-} mice after HSV-1 infection (Extended Data Fig. 4a). Besides, carvedilol didn't increase the production of *Ifnb*, *Isg15*, *Ccl5*, the phosphorylation of TBK1, IRF3, IRF7, STING in Arrb2^{-/-} peritoneal macrophages (Extended Data Fig. 4b-e). These results indicated that carvedilol promote type I IFN signaling through β -arrestin 2. Furthermore, the similar experiments in Figure 7 have been performed using Arrb2^{-/-}+control, Arrb2^{-/-}+WT, Arrb2^{-/-}+K171R and Arrb2^{-/-}+K171Q cells (Extended Data Fig. 5).

Question 7: Figure s3, the duplicate "inhibition of degradation" below in the diagram may be corrected as "promotion of deacetylation" according to the whole text.

Reply 7: We thank the reviewer for careful reading. We have corrected this expression.

Question 8: Check and correct English grammar and usage. For example, line 84, the first "and" was redundant; line 339-342, repetitive description.

Reply 8: We apologize about our mistake on English grammar. We have corrected our expression.

References:

1. Li, X. *et al.* Methyltransferase Dnmt3a upregulates HDAC9 to deacetylate the kinase TBK1 for activation of antiviral innate immunity. *NAT IMMUNOL* **17**, 806-815 (2016).
2. Wang, S. *et al.* YAP antagonizes innate antiviral immunity and is targeted for lysosomal degradation through IKK ϵ -mediated phosphorylation. *NAT IMMUNOL* **18**, 733-743 (2017).
3. Liu, B. *et al.* The ubiquitin E3 ligase TRIM31 promotes aggregation and activation of the

signaling adaptor MAVS through Lys63-linked polyubiquitination. *NAT IMMUNOL* **18**, 214-224 (2017).

REVIEWERS' COMMENTS

Reviewer #1 (Remarks to the Author):

The authors addressed my points and added the requested data. At this point, however, I cannot recommend publication yet:

1) The experimental section describing the EMSA is missing and the labeling of the plot is insufficient. How much cGAS is used, 10mg/ml? At these concentrations, cGAS typically shifts all DNA, so I wonder why so little of the probe is shifted. In addition, I am bit worried by the large amounts of beta-arrestin needed to see an enhancement of the interaction (>1mg/ml). I guess that speaks for a rather weak enhancement, but at least a control should be included that w/o cGAS, but in presence of the highest amount of b-arrestin used, no shift is seen.

Editorial/minor: There are numerous spelling and grammar errors, especially in the revised section that could be easily solved with the spell checking function of a word processor. Other sentences need attention:

"We found that 1-185 truncation of β -arrestin 2 and 213-522 truncation of cGAS were important for their interaction (Extended Data Fig. 2f, g)."

Since a "truncation" cannot be important for an interaction, perhaps reformulate to "We found, using truncated proteins, that sequence regions 1-185 ... were important for their interactions ..."

Reviewer #2 (Remarks to the Author):

The authors have adequately addressed this review's concerns and questions. I am satisfied with the revision of this work and would be happy to support the publication of this work.

First of all, we thank you all for your careful review. All your comments have been carefully revised and a new revised submission has been uploaded. We highlighted all our responses below in red.

Reviewer #1 (Remarks to the Author):

The authors addressed my points and added the requested data.

We thank the reviewer for careful reading and comments about our work.

At this point, however, I cannot recommend publication yet:

1) The experimental section describing the EMSA is missing and the labeling of the plot is insufficient. How much cGAS is used, 10mg/ml? At these concentrations, cGAS typically shifts all DNA, so I wonder why so little of the probe is shifted. In addition, I am bit worried by the large amounts of beta-arrestin needed to see an enhancement of the interaction (>1mg/ml). I guess that speaks for a rather weak enhancement, but at least a control should be included that w/o cGAS, but in presence of the highest amount of b-arrestin used, no shift is seen.

Reply 1: We thank the reviewer very much for the question about the EMSA experiment. Actually, we repeated this experiment for many times before with different concentration of cGAS (including the one we showed in the initial submission). In the EMSA experiment of the initial submission, it seem that we got the weak result (the weak binding of cGAS and probe, the weak enhancement of beta-arrestin 2), just because we used the freezing and thawing samples, in which the proteins were partially degraded to get the weak result as we see. Fortunately, we have many other repeated results. We have replaced this figure in the new submission, we can see that that cGAS could shift DNA at 2mg/ml and beta-arrestin 2 can obviously enhance the binding at 1mg/ml.

Editorial/minor: There are numerous spelling and grammar errors, especially in the revised section that could be easily solved with the spell checking function of a word processor. Other sentences need attention:

As per the reviewer's suggestion, we have checked the spelling and grammar errors and edited our manuscript accordingly.

"We found that 1-185 trunction of β -arrestin 2 and 213-522 trunction of cGAS were important for their interaction (Extended Data Fig. 2f, g)."

Since a "truncation" cannot be important for an interaction, perhaps reformulate to "We found, using truncated proteins, that sequence regions 1-185 ... were important for their interactions ..."

As per the reviewer's suggestion, we have revised this sentence.

Reviewer #2 (Remarks to the Author):

The authors have adequately addressed this review's concerns and questions. I am satisfied with the revision of this work and would be happy to support the publication of this work.

We thank the reviewer for careful reading and positive comments about our work. We will try our best to format the manuscript to meet Nature Communications publication requirement.